# Gradient Transformer: Learning to Generate Updates for LLMs

Binh-Nguyen Nguyen [* 1]  Khang Tran [* 1]  NhatHai Phan [1]  Issa Khalil [2]

## Abstract

Many organizations lack computational resources to fine-tune large language models (LLMs) on private (unshareable) data for better utility, while fine-tuning tiny language models (TinyLMs) alone performs poorly. To address this bottleneck, we propose a data-free knowledge distillation framework that generates LLM update vectors based on TinyLMs fine-tuned on private data. An update vector is a vector of parameter changes from an initial model to its fine-tuned version on a dataset, capturing the effect of cumulative gradient steps during fine-tuning. The key idea of our framework is a novel **Gradient Transformer** that transforms TinyLM's update vectors into LLM's update vectors. As derived from shadow datasets, GRAD-TRANSFORMER captures the correlation between TinyLM and LLM update vectors, enabling third-party providers to generate LLM update vectors given the organization's TinyLM update vectors without accessing the organization's private data. The framework supports multi-organization collaboration to jointly update LLMs, improving performance and cost-efficiency. Extensive experiments across language modeling and reasoning tasks show that GRAD-TRANSFORMER remarkably outperforms state-of-the-art knowledge distillation baselines, even under strict differential privacy protection.

## 1. Introduction

Large language models (LLMs) demonstrate superior performance on complex tasks across a wide range of domains, and have become a core component of modern AI systems (Hadi et al., 2023; Raiaan et al., 2024). However, the cost of fine-tuning an LLM on private data and tasks is unaf-

fordable for most organizations with limited resources (Shen et al., 2024; Guo et al., 2025). There are two solutions for this problem: (1) An organization can afford fine-tuning a TinyLM on their private data locally (Zhang et al., 2025a); and (2) An organization sends their private data to a third-party service provider's cloud, where the service provider fine-tunes an LLM on the shared data and then provides API services to the organization using the fine-tuned model (VM et al., 2024). Although cost-effective, the former solution with a TinyLM usually results in poor model performance. Meanwhile, organizations cannot share their private data with the service provider due to privacy and security constraints in sensitive real-world applications (Mbonihankuye et al., 2019; Voigt & Von dem Bussche, 2017).

**Motivation.** To address this problem, a growing line of work studies data-free knowledge distillation, where the student model does not access the data used to train the teacher (Qi et al., 2025; Wei et al., 2025). These approaches train a generative model to synthesize data samples mimicking the private data, guided by the teacher model, and subsequently fine-tune the student using the generated samples. However, these approaches are computationally expensive, require retraining a generator for each new teacher and a large-scale synthetic dataset to fine-tune the student LLM. Recent studies show that synthetic data mimicking private datasets can inadvertently leak sensitive information, causing serious privacy concerns (Zhang et al., 2022; Annamalai et al., 2024). Another line of work studies weak-to-strong knowledge distillation, where knowledge from a weaker (smaller) model is transferred to a stronger (larger) model (Shin et al., 2025; Yao et al., 2025). However, these methods require data sharing between teacher and student, violating the privacy constraints of the organizations. Therefore, it demands a novel mechanism enabling organizations to fine-tune LLMs on their data under privacy constraints. To address this problem, we propose the first data-free weak-to-strong knowledge distillation method that directly generates LLM update vectors based on TinyLM's update vectors fine-tuned on the organization's private data.

**Challenges.** Developing such a mechanism encounters the following critical challenges. First, the structural and capacity mismatch between TinyLMs and LLMs makes direct knowledge transfer difficult, as the teacher (TinyLM) is less expressive than the student (LLM). Second, the privacy con-

---

[*]Equal contribution [1]Department of Data Science, New Jersey Institute of Technology, Newark, NJ, USA [2]Qatar Computing Research Institute, HBKU, Doha, Qatar. Correspondence to: NhatHai Phan <phan@njit.edu>.

*Proceedings of the 43rd International Conference on Machine Learning*, Seoul, South Korea. PMLR 306, 2026. Copyright 2026 by the author(s).

straints on clients' training data rule out standard fine-tuning and distillation pipelines. Finally, the extremely high dimensionality of both the TinyLM and LLM parameter spaces demands a scalable, expressive transformation mechanism.

**Our Method.** To address these challenges, we develop a novel **Gradient Transformer** (GRAD-TRANSFORMER) that incorporates a Transformer-based encoder-decoder model (Chung et al., 2024) learning to generate update vectors for target LLMs based on fine-tuned TinyLM's update vectors. Its key idea is to segment the parameters of a fine-tuned TinyLM into a sequence of block-wise update vectors, each of which is projected into the same dimension hidden state, and autoregressively generate the corresponding update vectors for each block of the target LLM. To train the GRAD-TRANSFORMER, we curate a dataset of update vector tuples capturing the correlation between the TinyLM and the LLM fine-tuned on the same dataset. Training the GRAD-TRANSFORMER on this dataset enables a third-party service provider to directly generate update vectors for the target LLMs based on the TinyLMs fine-tuned on the client's private dataset without accessing the client's private data. Thereby, GRAD-TRANSFORMER offer a distinct capability to bypass the costly fine-tuning process of the LLMs on the clients' private data.

**Evaluation.** Extensive theoretical analysis on generalization and utility bounds and experiments on six benchmark datasets spanning diverse tasks show that: GRAD-TRANSFORMER consistently outperforms all state-of-the-art knowledge distillation baselines. For instance, GRAD-TRANSFORMER achieves an average PGR of 91.88% compared with 58.94% of the best-performing baseline across six datasets, registering an improvement of 55.89%. Thanks to extensive training on optimal LLM update vectors curated from a public dataset with a similar distribution to the private dataset, GRAD-TRANSFORMER retains highly competitive performance under strict differential privacy protection applied on the clients' fine-tuned TinyLMs.

**Contributions.** We make the following contributions:

- We propose the first data-free weak-to-strong knowledge distillation method that directly generates LLM update vectors from the fine-tuned TinyLM's update vectors, enabling privacy-preserving and efficient LLM fine-tuning without accessing private data.
- We propose GRAD-TRANSFORMER, a novel architecture that incorporates a Transformer-based encoder-decoder model that learns to generate update vectors for target LLMs from the update vectors of TinyLMs.
- We conduct a thorough theoretical analysis and extensive experiments on GRAD-TRANSFORMER to provide guidelines for its practical adoption, and highlight its superior performance, outperforming state-of-the-art knowledge distillation baselines.

## 2. Related Works

**Knowledge Distillation.** Knowledge distillation (Hinton et al., 2014) is an advanced method to transfer capabilities of a teacher model to a student model, categorizing in two lines of work: **(1) Strong-to-weak**, in which the teacher model is larger and more advanced than the student model; and **(2) Weak-to-strong**, in which the teacher model is smaller and less advanced than the student model (Burns et al., 2024). In weak-to-strong mechanisms, knowledge from weaker models is transferred to stronger models using their weak supervision to support super-alignment. (Burns et al., 2024) tried various strategies such as confidence loss and bootstrapping. Overlap density was proposed to explore data characteristics that lead to weak-to-strong knowledge distillation (Shin et al., 2025). Also, several works provided theoretical analysis for this line of work (Lang et al., 2024; Dong et al., 2025; Yao et al., 2025). Nevertheless, these works require sharing data between student and teacher models, violating the organization's privacy constraint.

**Data-Free Knowledge Distillation.** Data-free knowledge distillation is a subset of knowledge distillation that does not require using the original training data of the teacher model (Lopes et al., 2017). This research is relevant to our setting, where access to data is restricted by privacy constraints (Hu et al., 2024b). In these works, a generator generates synthetic samples guided by the teacher model to mimic the original training data, which is then used to train the student model by aligning its predictions with those of the teacher model (Tran et al., 2024; Liu et al., 2024; Wei et al., 2025; Qi et al., 2025). However, this pipeline poses the privacy risk of leaking identifiable information in the generated samples (Giomi et al., 2023; Hu et al., 2024a), which fundamentally contradicts the privacy motivation of data-free knowledge distillation. In contrast, GRAD-TRANSFORMER operates on the update vectors of the TinyLMs, enabling organizations to fine-tune the LLM without sharing their private data.

## 3. Problem Formulation

Let us consider a client possesses a private dataset $D = \{z_i\}_{i=1}^{m}$ of a downstream task $\tau$ with $z_i \in \mathcal{Z}$ is a data sample from the sample space $\mathcal{Z}$, $m$ is the total number of data samples in $D$. The client aims to fine-tune a target LLM, $\theta_T \in \Theta_T$, on the private data $D$ for deployment in their systems. Constrained by limited resources, instead of the LLM, the client fine-tunes a TinyLM $\theta_S$ from an initial $\theta_S^0$ on their private data $D$ using a learning method, denoted as $\mathcal{A}(D, \theta_S)$, which optimizes the following objective:

$$\theta_S^* = \arg\min_{\theta_S} \frac{1}{m} \sum_{i=1}^{m} \ell(z_i, \theta_S), \qquad (1)$$

where $\ell(\cdot, \cdot)$ is a non-negative loss function.

The client sends the update vector $\Delta\theta_S \stackrel{\text{def}}{=} \theta_S^* - \theta_S^0$ to a (third-party) service provider, who generates an update vector $\Delta\theta_T$ for the target LLM $\theta_T$ from the received $\Delta\theta_S$ through a GRAD-TRANSFORMER $\mathcal{M} : \Delta\theta_S \to \Delta\theta_T$ (Eq. 2). Given the generated $\Delta\theta_T$, the service provider updates the target LLM from its initial parameter $\theta_T^0$ (Eq. 3). We formulate the process of updating the target LLM as follows:

$$\Delta\theta_T = \mathcal{M}(\Delta\theta_S), \tag{2}$$
$$\hat{\theta}_T = \theta_T^0 + \Delta\theta_T. \tag{3}$$

The fine-tuned model $\theta_S^* = \mathcal{A}(D, \theta_S^0)$ encodes the knowledge derived from the dataset $D$ of the downstream task $\tau$. Therefore, the GRAD-TRANSFORMER plays the role of knowledge distillation from a fine-tuned TinyLLM to an LLM. This mechanism enables the client to achieve higher performance from the LLM on their local tasks without exposing the private data to a third party.

We generalize the setting to $N$ clients, where each client $i$ fine-tunes a TinyLM $\theta_{S,i}^*$ on a private dataset $D_i = \{z_j\}_{j=1}^{m_i}$. To update the target LLM, the service provider aggregates the TinyLMs' update vectors as $\Delta\theta_S = \texttt{pool}(\Delta\theta_{S,1}, \ldots, \Delta\theta_{S,N})$ where $\Delta\theta_{S,i} = \theta_{S,i}^* - \theta_{S,i}^0$, $\texttt{pool}(\cdot)$ is an aggregation function (e.g., average, summation). The targeted LLM can then be updated as in Eqs. (2) and (3). As a result, our mechanism enables joint training of the target LLM, significantly improving model performance and cost-effectiveness. In practice, the learning method $\mathcal{A}$ can be a privacy-preserving learning mechanism (Abadi et al., 2016; Gong et al., 2020) that provides rigorous privacy protection for clients' private datasets.

**GRAD-TRANSFORMER Objective.** The key objective is designing an effective GRAD-TRANSFORMER $\mathcal{M}$ to transform the $\Delta\theta_S$ to an update vector of the target LLM $\Delta\theta_T$. We directly extend this objective to support collaborative fine-tuning: Given a set of TinyLMs' update vectors $\{\Delta\theta_{S,i}\}_{i=1}^N$ and outputs $\Delta\theta_T$ to update the target LLM from its inital parameter $\theta_T^0$, such that it achieves high utility on the downstream task $\tau$ across all the clients' private datasets $\{D_i\}_{i=1}^N$, as follows:

$$\Delta\theta_T = \mathcal{M}\left(\{\Delta\theta_{S,i}\}_{i=1}^N\right) \text{ s.t. } \min_{\Delta\theta_T} \sum_{i=1}^N \sum_{j=1}^{m_i} \ell(z_j, \theta_T^0 + \Delta\theta_T).$$

**Technical Challenges.** Designing $\mathcal{M}$ faces several fundamental technical challenges, as follows:

**(1)** Since $\theta_S$ and $\theta_T$ consist of billion parameters, designing an appropriate architecture for $\mathcal{M}$ is non-trivial. For example, a naive modeling approach that concatenates all parameters of TinyLMs and LLMs' update vectors then applies a projection layer before passing them to encoders would require up to trillions of parameters, limiting the

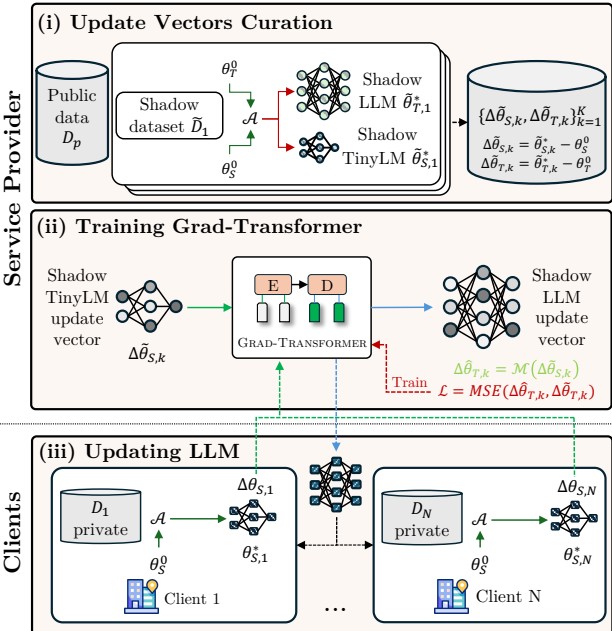

Figure 1. Overview of GRAD-TRANSFORMER's pipeline.

adoption of GRAD-TRANSFORMER in practice. Addressing this problem requires a novel architecture that scales efficiently with the vast parameter space of modern LLMs.

**(2)** The GRAD-TRANSFORMER $\mathcal{M}$ does not have access to clients' private datasets or their synthetic substitutions. This constraint prevents the service provider from distilling knowledge from TinyML's logits for specific data points and transferring it to the LLM, as in existing approaches. Bypassing this constraint requires rethinking the forms of knowledge that can be distilled and how to transfer them across models, rather than using logits.

## 4. Learning to Generate Updates for LLMs

To address these challenges, our learning framework consists of three phases (Figure 1): **(1)** Update vectors curation, **(2)** Training the GRAD-TRANSFORMER, and **(3)** LLM updating process, as follows.

### 4.1. Update Vectors Curation

To curate update vectors, the service provider collects a publicly available dataset $D_p$. By knowing the downstream task $\tau$, the dataset $D_p$ can be made more similar to clients' private datasets, thereby enhancing GRAD-TRANSFORMER's performance. When limited public datasets are available for task $\tau$, $D_p$ should cover as much general knowledge as possible to capture more correlation between the update vectors of TinyLMs and the LLMs. Then, it constructs a set of tuples $D_\Delta = \{(\Delta\tilde{\theta}_{S,k}, \Delta\tilde{\theta}_{T,k})\}_{k=1}^K$, where $|D_\Delta| = K$ is the number of tuples. These tuples in $D_\Delta$ capture the cross-model correlation between updated vectors, enabling

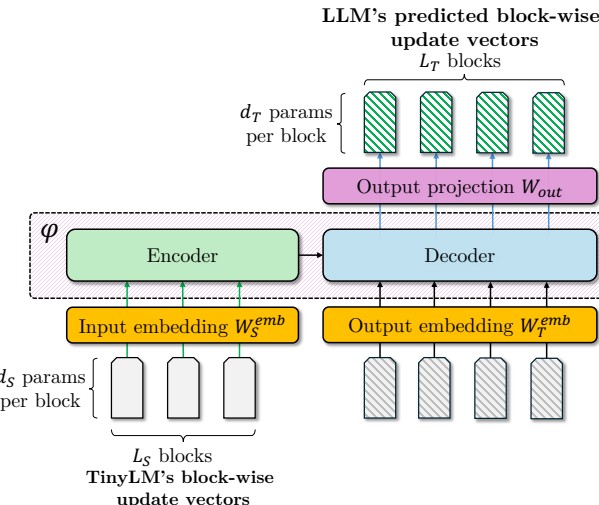

*Figure 2.* Overview of GRAD-TRANSFORMER Architecture.

learning to transform an update vector of the TinyLM to a corresponding update vector of the LLM.

Specifically, each tuple is derived as follows. Firstly, we randomly split $D_p$ into $K$ subsets $\{\tilde{D}_k\}_{k=1}^K$. Then, the initial TinyLM $\theta_S^0$ and LLM $\theta_T^0$ are fine-tuned on each shadow subset by the learning mechanism $\mathcal{A}$, producing the sets of shadow TinyLMs $\{\tilde{\theta}_{S,k}^*\}_{k=1}^K$ and shadow LLMs $\{\tilde{\theta}_{T,k}^*\}_{k=1}^K$. It is worth noting that the same initial parameters, $\theta_S^0$ and $\theta_T^0$, are used for each fine-tuning process, ensuring consistency in the root of the update vectors. Then, we compute $\Delta\tilde{\theta}_{S,k} = \tilde{\theta}_{S,k}^* - \theta_S^0$ and $\Delta\tilde{\theta}_{T,k} = \tilde{\theta}_{T,k}^* - \theta_T^0$. The key advantage of this design is low variance update vectors resulting in improving numerical stability. In addition, operating on update vectors supports low-rank approximation methods such as LoRA (Hu et al., 2022), enabling more flexible adoption of our mechanism.

### 4.2. GRAD-TRANSFORMER

Given the set of tuples $D_\Delta$, the service provider trains a GRAD-TRANSFORMER to take $\Delta\tilde{\theta}_{S,k}$ as input and generate $\Delta\tilde{\theta}_{T,k}$. To address the challenge posed by the large parameter space, we propose a novel GRAD-TRANSFORMER architecture that segments each update vector into a sequence of block-wise update vectors, each corresponding to a model's attention block (Figure 2). By projecting the block-wise update vector into a predefined dimension, our GRAD-TRANSFORMER enables the adoption of an encoder-decoder transformer taking a sequence of block-wise updates from a TinyLM to generate a sequence of block-wise updates of the LLM. The specific structure of our GRAD-TRANSFORMER is as follows.

**Model Architecture.** For every attention block in the TinyLM and LLM models, GRAD-TRANSFORMER concatenates the parameter weights in the update vector asso-

ciated with that block (including the query, key, value, and linear projection weights) into a single block-wise update vector. Now, the update vectors $\Delta\tilde{\theta}_{S,k}$ and $\Delta\tilde{\theta}_{T,k}$ can be decomposed as follows:

$$\Delta\tilde{\theta}_{S,k} = \{\delta_{S,k}^1, \ldots, \delta_{S,k}^{L_S}\}, \delta_{S,k}^j \in \mathbb{R}^{d_S}, \forall j \in [L_S], \quad (4)$$

$$\Delta\tilde{\theta}_{T,k} = \{\delta_{T,k}^1, \ldots, \delta_{T,k}^{L_T}\}, \delta_{T,k}^j \in \mathbb{R}^{d_T}, \forall j \in [L_T], \quad (5)$$

where $d_S$ and $d_T$ respectively are the numbers of concatenated parameters of an attention block of $\theta_S$ and $\theta_T$, $L_S$ and $L_T$ are the number of attention blocks in $\theta_S$ and $\theta_T$ correspondingly, and $\delta_{S,k}^j$ and $\delta_{T,k}^j$ are the block-wise update vectors from $\theta_S$ and $\theta_T$.

The block-wise update vector, i.e., $\delta_{S,k}^j$ and $\delta_{T,k}^j$, serves as a token-like unit in the input and output sequence processed by a transformer model (Yousuf et al., 2021). During the training process, each block-wise update vectors of the TinyLM model and LLM model are projected into hidden states by embedding layers $W_S^{emb}$ and $W_T^{emb}$, following the teacher-forcing paradigm (Williams & Zipser, 1989):

$$h_{S,k}^j = W_S^{emb}(\delta_{S,k}^j), \quad \forall j \in [L_S], \quad (6)$$

$$h_{T,k}^j = W_T^{emb}(\delta_{T,k}^j), \quad \forall j \in [L_T]. \quad (7)$$

Then, GRAD-TRANSFORMER adopts an encoder-decoder model $\varphi(\cdot)$ to generate the next block-wise update vectors of target LLMs in an auto-regressive manner (Graves, 2013). Specifically, at each decoding step, the model conditions on the encoded hidden states $\{h_{S,k}^1, \ldots, h_{S,k}^{L_S}\}$ and the hidden states of previous blocks in the target LLM to predict the hidden state of the next block, as follows:

$$\forall j \in [L_T] : \hat{h}_{T,k}^j = \varphi(h_{S,k}^1, \ldots, h_{S,k}^{L_S}, h_{T,k}^{<j}). \quad (8)$$

Finally, the hidden state $\hat{h}_{T,k}^j (\forall j \in [L_T])$ is projected back to block-wise update vectors through a layer $W_{out}(\cdot)$:

$$\forall j \in [L_T] : \hat{\delta}_{T,k}^j = W_{out}(\hat{h}_{T,k}^j), \quad (9)$$

where $\{\hat{\delta}_{T,k}^j\}_j$ serves as the prediction of the (ground-truth) LLM update vectors $\{\delta_{T,k}^j\}_j$.

**Training.** By minimizing a mean squared error loss between predicted update vectors $\{\hat{\delta}_{T,k}^j\}_j$ and ground-truth update vectors $\{\delta_{T,k}^j\}_j$, the service provider trains the GRAD-TRANSFORMER $\mathcal{M}$. Let $w \in \mathcal{W}$ is the trainable parameter of $\mathcal{M}$, the training objective is as follows:

$$\arg\min_{w\in\mathcal{W}} \frac{1}{K \times L_T} \sum_{k=1}^K \sum_{j=1}^{L_T} \|\hat{\delta}_{T,k}^j - \delta_{T,k}^j\|_2^2, \quad (10)$$

where $\|\cdot\|_2$ is the L2 norm of a vector. The parameters are trained on the set $D_\Delta$ using advanced gradient-based optimizers, such as Adam (Kingma, 2015).

The architecture and training of GRAD-TRANSFORMER are directly adaptable to the fully connected layers and embedding layers of the TinyLMs and LLMs, using customized $W_S^{emb}$, $W_T^{emb}$, and $W_{out}$ for those layers which align with their number of parameters. In addition, GRAD-TRANSFORMER is easily adaptable with Low-rank Approximation fine-tuning methods (Hu et al., 2022; Bałazy et al., 2025) to further enhance the scalability of our mechanism. In this setting, $\Delta\tilde{\theta}_S$ and $\Delta\tilde{\theta}_T$ are the fine-tuned LoRA adapters to shrink the parameter space of the update vectors.

### 4.3. Updating LLM

To update the target LLM, the service provider first sends the initial model parameters $\theta_S^0$ to all clients. For the $i$-th client, the initial model $\theta_S^0$ is fine-tuned using the local private dataset $D_i$ with learning method $\mathcal{A}$ to obtain the fine-tuned model $\theta_{S,i}^* = \mathcal{A}(D_i, \theta_S^0)$ and derive its update vector $\Delta\theta_{S,i} = \theta_{S,i}^* - \theta_S$. Then, each client sends its update vector to the service provider, which aggregates them to obtain $\Delta\theta_S = \frac{1}{N}\sum_{i=1}^{N}\Delta\theta_{S,i}$, where $N$ is the number of clients. Thereafter, $\Delta\theta_S$ is fed to the GRAD-TRANSFORMER $\mathcal{M}$, which outputs the predicted update vector of the target LLM $\Delta\hat{\theta}_T = \mathcal{M}(\Delta\theta_S)$. It iw worth noting that, to generate $\Delta\hat{\theta}_T$, the GRAD-TRANSFORMER recursively concatenates the predicted block-wise update vectors with the input hidden states and feeds them into $\varphi$ to generate the hidden states for the subsequent blocks:

$$\forall j \in [L_T]: \hat{h}_{T,i}^j = \varphi(h_{S,k}^1, \ldots, h_{S,k}^{L_S}, \hat{h}_{T,k}^{<j}), \quad (11)$$

and output the block-wise update vector as in Eq. (9). Finally, we add this predicted update vector to the original parameters of the target LLM to obtain the predicted target LLM $\hat{\theta}_T = \theta_T + \Delta\hat{\theta}_T$ and offer it to every client for inference or their deployments.

## 5. Theoretical Analysis

To provide a deeper understanding of the proposed method, we study GRAD-TRANSFORMER's generalization and utility bounds. We analyze the impact of the dataset $D_p$ and the learning mechanism $\mathcal{A}(\cdot)$ on the performance of $\mathcal{M}$ using an information-theoretic approach, shedding light on the key factors that affect its performance and providing suggestions for its practical adoption. Let's consider the downstream task $\tau$ inducing a probability distribution $\mu$ over the data sample space $\mathcal{Z}$. Similarly, the shadow dataset $D_p$ follows distribution $\tilde{\mu}$. Given a private dataset $D$, we define the empirical risk of $\mathcal{M}$ on $D$ as follows:

$$R_D(w) = \mathbb{E}_{\theta_S \sim P(\theta_S|D)}\Big[\sum_{i=1}^{N}\sum_{j=1}^{m_i}\ell(z_j, \theta_T^0 + \Delta\theta_T)\Big],$$

where the expectation is taken over the space of $\theta_S$ fine-tuned on $D$ by learning mechanism $\mathcal{A}$. Similarly, given a

distribution $\mu$ on the sample space $\mathcal{Z}$, we define the population risk with respect to $\mu$ of $\mathcal{M}$, as follows:

$$R_\mu(w) = \mathbb{E}_{\theta_S \sim P(\theta_S|D)}\Big[\mathbb{E}_{z \sim \mu}\ell(z, \theta_T^0 + \Delta\theta_T)\Big].$$

It is worth noting that we also adopt these empirical risk measures for the dataset $D_p$ and the distribution $\tilde{\mu}$.

We consider the Bayesian Network in Figure 3 for the training process of our mechanism, in which we can indicate that the parameter $w \in \mathcal{W}$ follows the conditional distribution

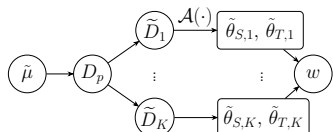

Figure 3. Bayesian Network of the GRAD-TRANSFORMER framework.

$P(w|D_p)$. In addition, we also consider an assumption that the loss function $\ell(\cdot)$ follows a $\sigma$-subgaussian distribution. This assumption is widely adopted for generalization bounds for knowledge distillation (Zhang et al., 2025b), foundation model analysis (Zhu et al., 2024), and other advanced learning methods such as meta learning (Chen et al., 2021a; Wu et al., 2024). From these considerations, we can derive the following generalization and utility analysis.

**Generalization Analysis.** We focus on understanding the generalization of $\mathcal{M}$ across the distribution $\tilde{\mu}$, defined by:

$$\text{Gen}(w) = R_{\tilde{\mu}}(w) - R_{D_p}(w). \quad (12)$$

The following lemma upper bounds this metric to derive a theoretical guarantee on the generalization of $\mathcal{M}$.

**Lemma 5.1.** *Suppose $\ell(\cdot, \cdot)$ follows a $\sigma$-subgaussian distribution. In expectation over parameter space $\mathcal{W}$ and the sampling process of $D_p$ from $\tilde{\mu}$, the generalization of $w$ is bounded as follows:*

$$|\mathbb{E}_{w, D_p}\text{Gen}(w)| \le 2\sqrt{\frac{\sigma^2 I(w; D_p)}{2|D_p|}}, \quad (13)$$

*where $I(w; D_p)$ is the mutual information between $w$ and $D_p$, and $|D_p|$ is the number of data samples in $D_p$.*

Lemma 5.1 suggests that the mutual information between $D_p$ and $w$ derived from $D_p$, which quantifies the dependence of $w$ on $D_p$, directly bounds the generalization of $\mathcal{M}$ on $\tilde{\mu}$. In addition, Lemma 5.1 also suggests that a larger $D_p$ leads to better generalization of $\mathcal{M}$, which is intuitive as a larger dataset $D_p$ allows $\mathcal{M}$ to better capture the distribution $\tilde{\mu}$, thereby reducing the mutual information between the trained $w$ and individual samples.

**Utility Bound.** We also provide utility bound for GRAD-TRANSFORMER framework at the inference time on a

client's private dataset $D$, which is measured by:

$$\texttt{Util}(w) = R_D(w) - R_{D_p}(w). \tag{14}$$

**Theorem 5.2.** *Suppose $\ell(\cdot, \cdot)$ follows a $\sigma$-subgaussian distribution. Then, in expectation over parameter space $\mathcal{W}$ and the sampling process of $D_p$ from $\tilde{\mu}$, the utility bound of $w$ on target dataset $D$ is bounded as follows:*

$$\left| \mathbb{E}_{w, D_p} \texttt{Util}(w) \right| \leq 2\sqrt{\frac{\sigma^2 [I(w; D_p) + KL(\tilde{\mu} \| \mu)]}{2|D|}},$$

*where $I(w; D_p)$ is the mutual information between $w$ and $D_p$, $KL(\cdot\|\cdot)$ is the KL-divergence, and $|D|$ is the size of $D$.*

Theorem 5.2 also indicates that the utility error bound is determined by the mutual information between $D_p$ and $w$, with the addition of the distributional distance between the shadow dataset and the client's private dataset. Consequently, the performance of $\mathcal{M}$ improves when $w$ depends less on $D_p$, and the shadow and private data distributions are more aligned. In addition, Theorem 5.2 also suggests that the utility bound decreases as the private dataset is larger, which is intuitive since the performance of $\theta_S$ fine-tuned on $D$ improves with data scale (Kaplan et al., 2020), resulting in better updated LLMs from $\theta_S$.

**Impact of Learning Mechanism $\mathcal{A}$.** Both Lemma 5.1 and Theorem 5.2 suggests that by controling the mutual information between $D_p$ and $w$, we can improve the generalization and utility of $\mathcal{M}$. To do so, we analyze the impact of $\mathcal{A}$ since $w$ is trained from update vectors derived from $\mathcal{A}$. From the Bayesian Network in Figure 3, we derive the following Corollary.

**Corollary 5.3.** *The utility bound and the generalization of $\mathcal{M}$ depends on the generalization of learning mechanism $\mathcal{A}$, and scale at a rate of $\mathcal{O}\left( \sqrt{\sum_{k=1}^{K} I(\tilde{\theta}_{S,k}, \tilde{\theta}_{T,k}; D_p)} \right)$ since $I(w; D_p) \leq \sum_{k=1}^{K} I(\tilde{\theta}_{S,k}, \tilde{\theta}_{T,k}; D_p)$.*

The proof of Corollary 5.3 is derived from the data processing inequality (Cover, 1999) based on the Bayesian Network in Figure 3. It implies that practitioners can control the mutual information between $\tilde{D}_p$ and $w$ through the learning mechanism $\mathcal{A}$. Specifically, practitioners can apply learning mechanisms that regularize on the mutual information $I(\tilde{\theta}_{S,k}, \tilde{\theta}_{T,k}; D_p), \forall k$, such as the Gibbs algorithm and noisy empirical risk minimization algorithm (Kuzborskij et al., 2019; Bu et al., 2022; Zhu & Bu, 2024) to reduce the dependence of $\tilde{\theta}_{S,k}$ and $\tilde{\theta}_{T,k}$ on $D_p$, resulting in a lower mutual information between $w$ and $D_p$. The full proofs of this section are in Appx. B.

## 6. Experiments and Results

Our extensive experiments across language modeling and reasoning tasks validate the effectiveness of GRAD-TRANSFORMER. Our framework surpasses SoTA baselines across single-client and multiple-client settings even under strict differential privacy protection.

**Datasets and Metrics.** We use six benchmark datasets: AQuA-RAT (Ling et al., 2017), GSM8K (Cobbe et al., 2021), CommonsenseQA (Talmor et al., 2019), DROP (Dua et al., 2019), SAMSum (Gliwa et al., 2019), and DialogSum (Chen et al., 2021b), covering math reasoning, commonsense reasoning, discrete reasoning, and dialogue summarization. Detailed description of datasets is in Appx. A.1.

AQuA-RAT, GSM8K, CommonsenseQA, DROP uses exact match accuracy as the evaluation metric, and SAMSum, DialogSum use ROUGE-1 (Lin, 2004) as the evaluation metric (Appx. A.1). Additionally, for each experiment, we use Performance Gap Recovered (PGR), a standard metric used to evaluate weak-to-strong distillation methods (Burns et al., 2024), calculated as: PGR $= \frac{\hat{P}_T - P_S}{P_T - P_S}$, where $\hat{P}_T$ is the performance of the updated LLM $\hat{\theta}_T$, $P_S$ and $P_T$ are the performance of the TinyLM and LLM models fine-tuned directly on the client's private dataset with mechanism $\mathcal{A}(\cdot)$, respectively. $P_T$ plays the role of the ceiling performance we can achieve on the private dataset.

**Constructing Update Vector Tuples $D_\Delta$.** For a considered dataset, we follow (Burns et al., 2024) to randomly split its training set into two parts: one is used as the client's private data $D$, and the other is used as public dataset $D_p$. We split the public dataset into $K = 300$ shadow datasets by randomly choosing 1,024 data samples for each shadow dataset. We employ a supervised fine-tuning mechanism $\mathcal{A}$ with LoRA (Hu et al., 2022) using rank $r = 2$, in which the LoRA adaptations are update vectors. For each shadow dataset, we fine-tune the models until they converge and collect LoRA adaptations of the TinyLM and the LLM models from 200 last training steps. Thereby, with each shadow dataset, we obtain 200 update vector tuples. In total, we have 60,000 update vector tuples to train the GRAD-TRANSFORMER. We randomly split these curated tuples into training and validation sets at a 95:5 ratio.

**Models.** We use the Qwen2.5 family models (Qwen et al., 2025) as the TinyLM and the LLM models. In particular, we use the Qwen2.5-3B-Instruct model as the TinyLM model and the Qwen2.5-7B-Instruct model as the LLM model. For clients, fine-tuning the 3B model TinyLM is significantly more efficient compared to the 7B LLM (Table 11, Appx. E). In addition, to analyze the scalability of our mechanism, we use TinyLM model scales from 0.5B to 3B, and LLM model scales from 7B to 14B. For GRAD-TRANSFORMER, we use Flan-T5-Large (Chung et al., 2024) as the Transformer-based encoder-decoder $\varphi$. To train GRAD-TRANSFORMER, we use batch size 32, learning rate ranging from 2e-5 to 8e-5 across datasets, and train for 30 epochs.

*Table 1.* Results in the Single Client setting on six different datasets. The best and second-best results are highlighted in **bold** and underline, respectively. Acc: Accuracy, R-1: ROUGE-1.

|  | Metric | $P_S$ | W2S | Conf | VisSup | Ours | $P_T$ |
|---|---|---|---|---|---|---|---|
| *Access client's private data* | – | – | ✓ | ✓ | ✓ | ✗ | – |
| AQuA-RAT | Acc | 48.43 | 43.70 | 47.64 | 47.24 | **61.02** | 58.66 |
|  | PGR | – | -46.24 | -7.72 | -11.63 | **123.07** | – |
| GSM8K | Acc | 62.62 | 71.72 | **74.30** | 72.71 | 73.59 | 73.16 |
|  | PGR | – | 86.34 | **110.82** | 95.73 | 104.08 | – |
| DROP | Acc | 49.36 | 51.18 | 54.18 | 51.87 | **58.26** | 59.01 |
|  | PGR | – | 18.86 | 49.95 | 26.01 | **92.23** | – |
| CQA | Acc | 77.40 | 82.64 | **83.46** | **83.46** | 83.21 | 83.78 |
|  | PGR | – | 82.13 | **94.98** | **94.98** | 91.07 | – |
| SAMSum | R-1 | 47.64 | 49.66 | 49.92 | 49.85 | **50.52** | 50.59 |
|  | PGR | – | 68.47 | 77.29 | 74.92 | **97.63** | – |
| DialogSum | R-1 | 46.43 | 47.00 | 47.70 | 46.68 | **48.37** | 50.92 |
|  | PGR | – | 12.69 | 28.29 | 5.57 | **43.21** | – |

*Table 2.* Results of different scales of TinyLM and LLM on AQuA-RAT dataset using accuracy evaluation metric.

|  | Client 1 | Client 2 | Client 3 | Client 4 | Client 5 | Avg. |
|---|---|---|---|---|---|---|
| $P_S$ (0.5B) | 29.13 | 27.49 | 30.15 | 31.08 | 27.49 | 29.07 |
| $P_S$ (1.5B) | 40.92 | 40.72 | 38.05 | 40.41 | 39.69 | 39.96 |
| $P_S$ (3B) | 53.95 | 54.26 | 53.64 | 54.15 | 54.26 | 54.05 |
| Ours (0.5B → 7B) | 63.08 | 61.13 | 62.97 | 65.74 | 63.49 | 63.28 |
| Ours (1.5B → 7B) | 59.59 | 64.41 | 63.49 | 64.51 | 65.33 | 63.47 |
| Ours (3B → 7B) | 61.74 | 64.31 | 63.90 | 62.87 | 65.03 | 63.57 |
| $P_T$ (7B) | 64.92 | 66.05 | 64.51 | 64.21 | 66.46 | 65.23 |
| Ours (0.5B → 14B) | 68.00 | 65.54 | 67.08 | 69.13 | 68.41 | 67.63 |
| Ours (1.5B → 14B) | 67.08 | 65.13 | 67.28 | 70.67 | 68.21 | 67.67 |
| Ours (3B → 14B) | 68.41 | 65.85 | 67.59 | 69.33 | 67.49 | 67.73 |
| $P_T$ (14B) | 68.10 | 66.77 | 67.08 | 69.13 | 69.13 | 68.04 |

*Table 3.* Results in the Multiple Client setting. We report the result for each client, and the averaged result across 5 clients. The first six columns report accuracy for AQuA-RAT, CommonsenseQA, DROP and ROUGE-1 for SAMSum. The best and second-best results are highlighted in **bold** and underline, respectively.

|  | Client 1 | Client 2 | Client 3 | Client 4 | Client 5 | Avg. | Avg. PGR |
|---|---|---|---|---|---|---|---|
| **AQuA-RAT** | | | | | | | |
| $P_S$ | 53.95 | 54.26 | 53.64 | 54.15 | 54.26 | 54.05 | – |
| W2S | 52.31 | 55.18 | 56.10 | 59.28 | 56.00 | 55.77 | 15.38 |
| Conf | 55.38 | 58.15 | 56.82 | 59.79 | 59.49 | 57.93 | 34.70 |
| VisSup | 54.56 | 57.85 | 57.33 | 59.18 | 59.08 | 57.60 | 31.75 |
| **Ours** | **61.74** | **64.31** | **63.90** | **62.87** | **65.03** | **63.57** | **85.15** |
| $P_T$ | 64.92 | 66.05 | 64.51 | 64.21 | 66.46 | 65.23 | – |
| **CommonsenseQA** | | | | | | | |
| $P_S$ | 77.55 | 78.57 | 73.47 | 76.53 | 78.57 | 76.94 | – |
| W2S | 78.57 | 84.69 | 77.55 | **82.65** | 78.57 | 80.41 | 68.04 |
| Conf | 78.57 | **85.71** | 75.51 | 80.61 | **80.61** | 80.20 | 63.92 |
| VisSup | 78.57 | **85.71** | 76.53 | 81.63 | 79.59 | 80.41 | 68.04 |
| **Ours** | **80.61** | 83.67 | **83.67** | 79.59 | **80.61** | **81.63** | **91.96** |
| $P_T$ | 80.61 | 86.73 | 81.63 | 81.63 | 79.59 | 82.04 | – |
| **DROP** | | | | | | | |
| $P_S$ | 56.98 | 55.81 | 53.36 | 58.91 | 54.91 | 55.99 | - |
| W2S | 57.24 | 61.24 | 61.63 | 63.57 | 61.63 | 61.06 | 63.85 |
| Conf | 58.91 | 64.21 | **63.82** | 64.34 | 61.63 | 62.58 | 83.00 |
| VisSup | 56.85 | 63.44 | 62.14 | 63.82 | 61.11 | 61.47 | 69.02 |
| **Ours** | **60.72** | **64.34** | 61.37 | **64.47** | **62.53** | **62.69** | **84.38** |
| $P_T$ | 64.73 | 64.08 | 60.98 | 63.82 | 66.02 | 63.93 | – |
| **SAMSum** | | | | | | | |
| $P_S$ | 45.32 | 47.70 | 45.31 | 46.12 | 47.25 | 46.34 | – |
| W2S | 48.02 | 49.26 | 50.11 | 48.06 | 46.39 | 48.37 | 59.71 |
| Conf | **49.99** | 50.09 | **51.98** | 48.19 | 48.17 | **49.68** | **98.24** |
| VisSup | 48.51 | **50.60** | 50.35 | 49.14 | 47.47 | 49.21 | 84.41 |
| **Ours** | 47.92 | 50.50 | 48.83 | **49.37** | **48.45** | 49.01 | 78.53 |
| $P_T$ | 49.51 | 49.98 | 48.91 | 50.21 | 50.11 | 49.74 | – |

**Baselines.** We employ SoTA weak-to-strong distillation methods as baselines: **(1) W2S** (Burns et al., 2024), **(2) Conf** (Burns et al., 2024), **(3) VisSup** (Guo et al., 2024). We also compare our method to **(4)** $P_S$, which is the TinyLM fine-tuned from the client side, and **(5)** $P_T$ (*ceiling performance*), which is the performance when we fine-tune the LLM directly on the client's data. Prior data-free knowledge distillation methods are not applicable to our experimental setting, as they are designed for image or text classification and do not extend to the complex text generation tasks considered in our experiments. Detailed description of each baseline is in Appx. C.2. We also split the training data of considered benchmark datasets into two parts: one to train weak models, and the other for the distillation process.

**Performance for a Single Client.** In this experiment, we evaluate the performance of GRAD-TRANSFORMER in generating the LLM update using a single fine-tuned TinyLM. Table 1 shows that the generated LLM update vectors enables the updated LLM to outperform the fine-tuned TinyLM in terms of performance across all tasks. Also, GRAD-TRANSFORMER achieves an average PGR of 91.88% compared with 58.94% of the best-performing baseline (i.e., Conf) registering an improvement of 55.89% across all the tasks. It is worth noting that GRAD-TRANSFORMER does not need to access the client's private data compared with the baselines, highlighting its practicability in real-world applications. On some datasets, baselines that directly leverage client model's logits achieve higher performance, since GRAD-TRANSFORMER only uses on update vectors to preserve data privacy and fail to preserve task-specific information that is more explicitly encoded in model logits.

**Performance on Multiple Clients.** We focus on evaluating the performance of GRAD-TRANSFORMER with five different clients. In this setting, the first half that is used as client's private data is split into 5 different subsets, one for each client. Each subset is further split into training and test sets with a 90:10 ratio. Regarding $P_T$, we fine-tune the LLM on the combined data from all five clients and report its performance on the test set of each client. Table 3 shows that the average results of GRAD-TRANSFORMER consistently outperform all baselines, achieving an average PGR of 85.01% compared with 69.97% of the best-performing baseline (i.e., Conf) across four datasets. In addition, the generated LLM consistently surpasses the clients' TinyLMs $P_S$ across all datasets, providing markedly improved inference for the clients. This results strengthen the effectiveness of GRAD-TRANSFORMER given multiple clients.

**GRAD-TRANSFORMER with Differential Privacy (DP).**

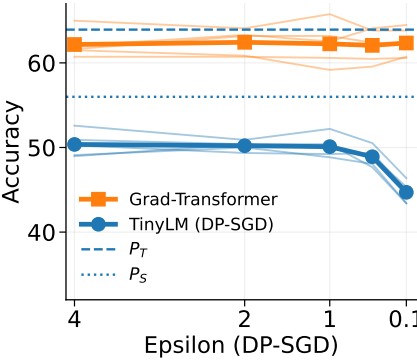

*Figure 4.* Averaged performance of GRAD-TRANSFORMER in the 5-client setting with DP-SGD training for clients on the DROP dataset. The light-colored lines show performance in each client.

*Table 4.* Results of different block-wise segmentation strategies on the AQuA-RAT dataset using accuracy evaluation metric.

|        | Client 1 | Client 2 | Client 3 | Client 4 | Client 5 | Avg. |
|--------|----------|----------|----------|----------|----------|------|
| $P_S$  | 53.95 | 54.26 | 53.64 | 54.15 | 54.26 | 54.05 |
| Q-V separately | 63.18 | 62.56 | 63.90 | 64.31 | 64.12 | 63.61 |
| Reverse block order | 38.36 | 37.13 | 39.49 | 40.92 | 42.77 | 39.73 |
| Ours   | 61.74 | 64.31 | 63.90 | 62.87 | 65.03 | 63.57 |
| $P_T$  | 64.92 | 66.05 | 64.51 | 64.21 | 66.46 | 65.23 |

In this experiment, we adopt DP-SGD (Abadi et al., 2016) as the client-side training mechanism $\mathcal{A}$ to provide rigorous DP protection for clients' private datasets. The privacy budget ranges from $\varepsilon = 4.0$ to $\varepsilon = 0.1$, guaranteeing a very strict privacy protection, and a broken probability of $\delta = 10^{-5}$, resulting in an $(\varepsilon, \delta)$-DP-preserving learning mechanism for every client. Figure 4 shows that on the DROP dataset, at $\varepsilon = 4.0$, the performance of the fine-tuned TinyLM and the generated LLM only drops marginally compared to the models from vanilla fine-tuning. In contrast, at a more strict privacy guarantee level (i.e., $\varepsilon = 0.1$), the performance of the TinyLM drops significantly (from 55.59% to 44.70%). However, GRAD-TRANSFORMER maintains a high performance even with noisy DP-SGD update vectors from the TinyLM, i.e., 62.35%. This consistency is achieved by extensively training GRAD-TRANSFORMER to generate optimal LLM update vectors, curated from $D_p$, which has a similar distribution to the private dataset, thereby enabling GRAD-TRANSFORMER to be robust against DP-preserving TinyLM's update vector. Results in other datasets, i.e., AQuA-RAT, CommonsenseQA (Figure 7, Appx. E), strengthen our observation by showing that GRAD-TRANSFORMER consistently performs well (88.02% PGR on average) under DP-SGD fine-tuning of the TinyLMs with $\varepsilon = 0.1$.

**Model Scales.** We explore the effectiveness of GRAD-TRANSFORMER in scaling the size difference between the TinyLM and the LLM in the following experiments (Table 2): **(1)** We fix the LLM to be 7B model, and reduce the

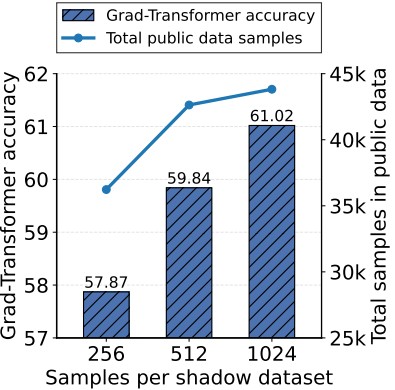

*Figure 5.* Ablation study on the number of samples in a shadow dataset and GRAD-TRANSFORMER's performance on AQuA-RAT.

size of the TinyLM from 3B from previous experiments to 0.5B; and **(2)** Given these varying TinyLMs, we increase the size of the LLM from 7B from previous experiments to 14B. In the first experiment, reducing the size of the TinyLM from 3B to 0.5B does not affect the performance of GRAD-TRANSFORMER in generating the LLM update vectors. In fact, the LLM, i.e., 7B, still perform competitively given different sizes, i.e., 0.5B, 1.5B, 3B, of the TinyLM. In the second experiment, given these sizes of TinyLMs, the GRAD-TRANSFORMER can still effectively generate the LLM update vectors with similar performance to the target 14B LLM. Although not effective in scaling the LLM to a 32B model given a 0.5B TinyLM given the current limited size of the GRAD-TRANSFORMER (Flan-T5-Large as Transformer encoder-decoder model $\varphi$), these promising results show that GRAD-TRANSFORMER can work effectively even with a $28\times$ scale up in model size (i.e., 0.5B TinyLM to 14B LLM).

**Update Vector Partitioning Strategies.** In Table 4, we explore and show the results of alternative update vector partitioning strategies in GRAD-TRANSFORMER other than our proposed block-wise segmentation. We experiment on two different strategies: **(1) Q-V separately** splits the update vectors by weight type as a finer grain level in addition to the block-wise partition as follows: One GRAD-TRANSFORMER is trained on only the block-wise query weight updates, and a second GRAD-TRANSFORMER is trained on only the block-wise value weight updates. Afterwards, the outputs from these two GRAD-TRANSFORMERs are merged to form the update vector for the LLM. Note that in the proposed method, we concatenate query and value weight updates of each block to train a single GRAD-TRANSFORMER. This method gives similar performance to our proposed method (i.e., only a 0.04 difference in performance). However, this method is not scalable and takes up significantly more computing since we need to train multiple GRAD-TRANSFORMERs for each small component of each block. **(2) Reverse block order** reverses the block

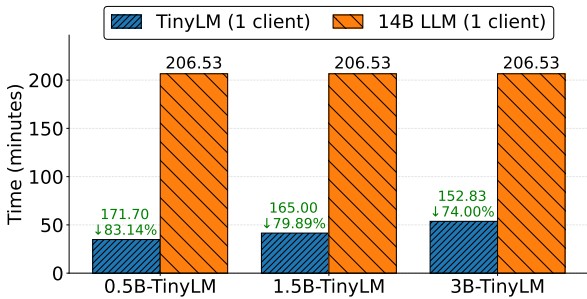

*Figure 6.* Time consumption (minutes) when fine-tuning 0.5B, 1.5B or 3B TinyLM compared to fine-tuning a 14B LLM for 1 client on AQuA-RAT dataset in the experiments in Table 2.

ordering of the weight update vectors before feeding them to GRAD-TRANSFORMER. This reversed block ordering disrupts the natural sequential dependency between layers. Performance drops further to 40.31%, confirming that the autoregressive decoder relies on the correct layer ordering to capture inter-block correlations. These results demonstrate that our block-wise segmentation strategy, which preserves both intra-block weight correlations and the inter-block sequential structure, is critical to GRAD-TRANSFORMER's effectiveness.

**Shadow Dataset.** Figure 5 reports the performance of GRAD-TRANSFORMER on the AQuA-RAT dataset as the number of samples in each shadow dataset $\tilde{D}_i$ varies. We can observe that the performance of GRAD-TRANSFORMER increases monotonically with the number of samples per shadow dataset and the total number of samples in the public dataset $D_p$. This result is consistent with our theoretical analysis (Lemma 5.1). Performance is highest at 1,024 samples (i.e., 61.02%).

**Update Vector Tuples** $D_\Delta$. Figure 8 (Appx E) shows the performance of GRAD-TRANSFORMER on AQuA-RAT when using different numbers of update tuples. By varying the number of update vector tuples during training, we examine how many are needed for the GRAD-TRANSFORMER to reach its optimal performance. We observe that at 1,000 update vector tuples, the performance of GRAD-TRANSFORMER has converged to its optimal value.

**Time Consumption.** We explore the time efficiency that GRAD-TRANSFORMER offers for clients. All time consumption analysis are conducted with $1\times$ NVIDIA A100 80GB GPU. Using GRAD-TRANSFORMER to update a 14B LLM from update vectors of a 0.5B TinyLM (as in the experiments in Table 2) provides remarkable efficiency gains compared to directly fine-tuning the 14B LLM (Figure 6). In fact, the fine-tuning time is reduced by 83.14%, representing a substantial efficiency gain for clients. Note that the training cost of the GRAD-TRANSFORMER is marginal since it only needs 1,000 update tuples to reach optimal performance (Figure 8, Appx. E).

## 7. Conclusions

In this paper, we propose a new framework to generate update vectors of LLMs based on update vectors of TinyLMs fine-tuned on private data. To achieve our goal, we develop a novel GRAD-TRANSFORMER architecture trained on update vectors derived by fine-tuning TinyLMs and LLMs on public shadow datasets. GRAD-TRANSFORMER enables multi-organization collaboration to jointly update LLMs, improving performance and cost-efficiency. Extensive experiments show that GRAD-TRANSFORMER outperforms SoTA knowledge distillation baselines, even under strict differential privacy protection.

## Impact Statement

In developing GRAD-TRANSFORMER, no human subjects were involved, so there are no ethical concerns related to data privacy. This paper provides a framework to support multi-organization collaboration to jointly update LLMs. This framework preserves privacy by ensuring that all private data remains on clients' local devices, eliminating the need to share it with external servers. Our work contributes to the development of secure and trustworthy AI, ensuring a safer, more efficient pipeline for fine-tuning LLMs from TinyLMs.

## Reproducibility Statement

We have provided sufficient implementations details in the main paper and the appendix to ensure reproducibility of the results presented in this paper. Our code is available at: https://github.com/nguyenrtm/grad-transformer

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

# A. Datasets and Evaluation Metrics

## A.1. Dataset descriptions

*Table 5.* Datasets, tasks, and evaluation metrics in experiments.

| Dataset | Task | Metric |
|---|---|---|
| AQuA-RAT | Math Reasoning | Acc (EM) |
| GSM8K | Math Reasoning | Acc (EM) |
| CommonsenseQA | Commonsense Reasoning | Acc (EM) |
| DROP | Discrete Reasoning | Acc (EM) |
| SAMSum | Dialogue Summarization | ROUGE-1 |
| DialogSum | Dialogue Summarization | ROUGE-1 |

*Acc: Accuracy, EM: Exact Match*

The detailed descriptions of six datasets used in our experiments are below:

- **AQuA-RAT** (Algebra Question Answering with Rationales) (Ling et al., 2017): AQuA-RAT contains algebraic word problems paired with natural language rationales. Each instance includes a problem description in natural language and five multiple-choice options, exactly one of which is correct. The ground-truth annotation provides both a textual explanation of the solution and the correct answer choice. The dataset comprises 97.5k training samples and 254 validation samples.

- **GSM8K** (Grade School Math 8K) (Cobbe et al., 2021): GSM8K is a collection of high-quality, linguistically diverse grade-school mathematics word problems designed to evaluate question answering with multi-step numerical reasoning. Each problem typically requires between two and eight reasoning steps and is solved through a sequence of elementary arithmetic operations. Solutions are expressed in natural language. The dataset contains 7.5k training samples and 1.3k test samples.

- **CommonsenseQA** (Talmor et al., 2019): CommonsenseQA is a multiple-choice question answering dataset that requires reasoning over diverse types of commonsense knowledge. Each question is paired with five answer options, exactly one of which is correct. The dataset consists of 9.7k training samples and 1.2k validation samples.

- **DROP** (Discrete Reasoning Over Paragraphs) (Dua et al., 2019): DROP is a crowdsourced, adversarially constructed reading comprehension dataset that requires models to resolve references within a question, potentially across multiple spans in the input paragraph, and to perform discrete reasoning operations such as addition, counting, and sorting. These operations demand deeper semantic understanding of paragraph content than earlier reading comprehension benchmarks. The dataset contains 77.4k training samples and 9.54k validation samples.

- **SAMSum** (Gliwa et al., 2019): SAMSum is a dialogue summarization dataset consisting of messenger-style conversations paired with human-written summaries. The conversations were authored by linguists fluent in English to reflect everyday messaging behavior and cover a realistic distribution of topics. The language style varies from informal to formal and may include slang, emoticons, and typographical errors. Each dialogue is annotated with a concise third-person summary capturing the main content of the conversation. The dataset includes 14.7k training samples and 818 validation samples.

- **DialogSum** (Chen et al., 2021b): DialogSum is a large-scale dialogue summarization dataset comprising multi-turn conversations annotated with manually written summaries and topic labels. The dataset contains 12.5k training samples and 500 validation samples.

## A.2. Evaluation metrics

For reasoning and question answering datasets AQuA-RAT, GSM8K, CommonsenseQA, DROP, we use exact match accuracy as the evaluation metric. Precisely, we prompt the model to generate the final answer at the end after its reasoning, and we compare this answer to the ground truth. If this generated answer exactly matches the ground truth label, we count it as the correct answer.

For dialogue summarization datasets SAMSum and DialogSum, we use Recall-Oriented Understudy for Gisting Evaluation score (ROUGE score) (Lin, 2004), which is a standard evaluation metric for summarization tasks. Specifically, we use ROUGE-1. The metric compares the generated summaries to a ground truth summaries, the higher scores show closer overlapping between them.

Additionally, for each experiment, we use performance gap recovered (PGR), a standard metric used to evaluate weak-to-strong distillation methods (Burns et al., 2024), as described in the paper.

## B. Theoretical Analysis

**Setting.** We consider a single down-stream task $\tau$ across $N$ clients' datasets, which induce a probability distribution $\mu$ over the data sample space $\mathcal{Z}$, i.e., $\forall i \in [N], \forall z \in D_i : z \sim \mu$. Similarly, the shadow dataset $D_p$ is equipped with a distribution $\tilde{\mu}$. For each subset $\tilde{D}_k \subset D_p$, we assume that each sample $z \in \tilde{D}_k$ is drawn i.i.d from the distribution $\tilde{\mu}$. Given a subset $\tilde{D}_k$, we denote $\phi_k = (\tilde{\theta}^*_{S,k}, \tilde{\theta}^*_{T,k})$ as the tuple of shadow models trained from $\tilde{D}_k$ using the learning mechanism $\mathcal{A}$, i.e., $\tilde{\theta}^*_{S,k} = \mathcal{A}(\tilde{D}_k, \theta^0_S)$ and $\tilde{\theta}^*_{T,k} = \mathcal{A}(\tilde{D}_k, \theta^0_T)$. It's worth noting that the learning mechanism $\mathcal{A}$ is typically a stochastic learning mechanism, resulting in the fact that $\tilde{\theta}^*_{S,k}$ and $\tilde{\theta}^*_{T,k}$ follow the conditional distributions $P(\theta_S|\tilde{D}_k)$ and $P(\theta_T|\tilde{D}_k)$, respectively.

### B.1. Proof of Lemma 5.1

We are interested in the generalization of $\mathcal{M}$ across the distribution $\mu$ which is defined by:

$$\text{Gen}(w) = R_{\tilde{\mu}}(w) - R_{D_p}(w), \tag{15}$$

namely the difference between the empirical risk on the shadow dataset $D_p$ and the population risk of $\mathcal{M}$ under the distribution $\tilde{\mu}$. We seek an upper bound for this metric to derive a theoretical guarantee on the generalization of $\mathcal{M}$, from which we can extend to the client's distribution $\mu$ based on the divergence between $\tilde{\mu}$ and $\mu$.

**Lemma B.1.** *Suppose $\ell(\cdot, \cdot)$ follows a $\sigma$-subgaussian distribution. Then, in expectation over parameter space $\mathcal{W}$ and the sampling process of $D_p$ from $\tilde{\mu}$, the generalization of $w$ is bounded as follows:*

$$|\mathbb{E}_{w,D_p} \text{Gen}(w)| \leq 2\sqrt{\frac{\sigma^2 I(w, D_p)}{2|D_p|}}. \tag{16}$$

*Proof.* Since $w$ is trained on $D_p$, $w$ is dependent on $D_p$, i.e., $w \sim P(w|D_p)$. We denote $\tilde{w} \in \mathcal{W}$ as a copy of $w$ such that $\tilde{w}$ is independent from $D_p$ by marginalizing over $D_p$, i.e., $\tilde{w} \sim \mathbb{E}_{D_p} P(w|D_p)$. We define for any $\lambda \in \mathbb{R}$:

$$\psi_{\tilde{w},D_p}(\lambda) = \log \mathbb{E}_{\tilde{w},D_p}\left[e^{\lambda\left(R_{D_p}(\tilde{w}) - \mathbb{E}_{\tilde{w},D_p}[R_{D_p}(\tilde{w})]\right)}\right] \tag{17}$$

$$= \log \mathbb{E}_{\tilde{w},D_p}\left[e^{\lambda R_{D_p}(\tilde{w})}\right] - \lambda \mathbb{E}_{\tilde{w},D_p}[R_{D_p}(\tilde{w})]. \tag{18}$$

By deriving the mutual information between $w$ and $D_p$ using Donsker-Varadhan representation (Boucheron et al., 2013) with a measurable function $g$, we have:

$$I(w, D_p) = D_{KL}(P_{w,D_p} \| P_w P_{D_p}) \tag{19}$$

$$= \sup_g \left\{ \mathbb{E}_{w,D_p}[g(w, D_p)] - \log \mathbb{E}_{\tilde{w},D_p}\left[e^{g(\tilde{w},D_p)}\right] \right\} \tag{20}$$

$$\geq \lambda \mathbb{E}_{w,D_p}[R_{D_p}(w)] - \log \mathbb{E}_{\tilde{w},D_p}[e^{\lambda R_{D_p}(\tilde{w})}], \forall \lambda \in \mathbb{R} \tag{21}$$

$$= \lambda \mathbb{E}_{w,D_p}[R_{D_p}(w)] - \lambda \mathbb{E}_{\tilde{w},D_p}[R_{D_p}(\tilde{w})] - \psi_{\tilde{w},D_p}(\lambda). \tag{22}$$

It's worth noting that $D_p$ is i.i.d sampled from $\tilde{\mu}$, i.e., $\forall z \in D_p : z \sim \tilde{\mu}$, resulting in $D_p$ also follows the distribution $\tilde{\mu}$. We

then analyze $\lambda\mathbb{E}_{\tilde{w},D_p}[R_{D_p(\tilde{w})}]$, as follows:

$$\lambda\mathbb{E}_{\tilde{w},D_p}[R_{D_p(\tilde{w})}] = \lambda\mathbb{E}_{\tilde{w}}\mathbb{E}_{D_p}[R_{D_p}(\tilde{w})] = \lambda\mathbb{E}_{\tilde{w}}\Big[\frac{1}{\tilde{m}}\sum_{i=1}^{\tilde{m}}\mathbb{E}_{D_p}\ell(\tilde{\theta}_T,z_i)\Big] \tag{23}$$

$$= \lambda\mathbb{E}_{\tilde{w}}\Big[\frac{1}{\tilde{m}}\sum_{i=1}^{\tilde{m}}\mathbb{E}_{z_i\sim\tilde{\mu}}\ell(\tilde{\theta}_T,z_i)\Big] = \lambda\mathbb{E}_{\tilde{w}}[R_{\tilde{\mu}}(\tilde{w})] \tag{24}$$

$$= \lambda\mathbb{E}_{w,D_p}[R_{\tilde{\mu}}(w)], \tag{25}$$

where the last equation is conditionalized on the distribution of $\tilde{w}$, i.e., $\tilde{w} \sim \mathbb{E}_{D_p}P(w|D_p)$. If we adopt Eq.(25) into Eq.(22), we have that:

$$-\lambda\Big(\mathbb{E}_{w,D_p}[R_{\tilde{\mu}}(w)] - \mathbb{E}_{w,D_p}[R_{D_p}(w)]\Big) \le I(w,D_p) - \psi_{\tilde{w},D_p}(\lambda), \quad \forall\lambda\in\mathbb{R} \tag{26}$$

$$\mathbb{E}_{w,D_p}[R_{\tilde{\mu}}(w)] - \mathbb{E}_{w,D_p}[R_{D_p}(w)] \le \frac{I(w,D_p) + \psi_{\tilde{w},D_p}(-\lambda)}{\lambda}, \quad \forall\lambda>0 \tag{27}$$

By the assumption that $\ell(\cdot,\cdot)$ follows a $\sigma$-subgaussian, we can have that $R_{D_p}(\tilde{w})$ follows $\frac{\sigma}{\sqrt{|D_p|}}$-subgaussian. Since $\psi_{\tilde{w},D_p}(\lambda)$ is the cummulant generative function of random variable $R_{D_p}(\tilde{w}) - \mathbb{E}_{\tilde{w},D_p}[R_{D_p}(\tilde{w})]$, which is also $\frac{\sigma}{\sqrt{|D_p|}}$-subgaussian due to linearity, we have that:

$$\psi_{\tilde{w},D_p}(\lambda) \le \frac{\lambda^2\sigma^2}{2|D_p|}, \quad \forall\lambda\in\mathbb{R}. \tag{28}$$

Adopting Eq.(28), we have that:

$$\mathbb{E}_{w,D_p}[R_{\tilde{\mu}}(w)] - \mathbb{E}_{w,D_p}[R_{D_p}(w)] \le \frac{I(w,D_p)}{\lambda} + \frac{\lambda\sigma^2}{2|D_p|}, \quad \forall\lambda>0. \tag{29}$$

By applying Cauchy-Schwarz inequality, we can choose $\lambda > 0$ that minimizes the right-hand side of the equation. Therefore,

$$\mathbb{E}_{w,D_p}\Big[R_{\tilde{\mu}}(w)\Big] - R_{D_p}(w) \le 2\sqrt{\frac{\sigma^2 I(w,D_p)}{2|D_p|}}. \tag{30}$$

If we choose $\lambda < 0$ in Eq. (27), similar analysis results in:

$$\mathbb{E}_{w,D_p}\Big[R_{\tilde{\mu}}(w)\Big] - R_{D_p}(w) \ge -2\sqrt{\frac{\sigma^2 I(w,D_p)}{2|D_p|}}. \tag{31}$$

Thus, we have that:

$$\Big|\mathbb{E}_{w,D_p}\Big[R_{\tilde{\mu}}(w) - R_{D_p}(w)\Big]\Big| \le 2\sqrt{\frac{\sigma^2 I(w,D_p)}{2|D_p|}}, \tag{32}$$

which concludes the proof. □

### B.2. Proof of Theorem 5.2

**Theorem B.2.** *Suppose $\ell(\cdot,\cdot)$ follows a $\sigma$-subgaussian distribution. Then, in expectation over parameter space $\mathcal{W}$ and the sampling process of $D_p$ from $\tilde{\mu}$, the utility bound of $w$ on target dataset $D$ is bounded as follows:*

$$\Big|\mathbb{E}_{w,D_p}\Big[R_D(w) - R_{D_p}(w)\Big]\Big| \le 2\sqrt{\frac{\sigma^2[I(w,D_p) + KL(\tilde{\mu}\|\mu)]}{2|D|}}. \tag{33}$$

*Proof.* To start with the proof, we consider the following KL-divergence:

$$KL(P_{w,D_p}\|P_w \otimes \mu) = I(w, D_p) + KL(P_{D_p}\|\mu) \tag{34}$$

$$= I(w, D_p) + KL(\tilde{\mu}\|\mu), \tag{35}$$

where the second equation is from the fact that $D_p \sim \tilde{\mu}$. Similar to the proof of Lemma B.1, we denote $\tilde{w} \in \mathcal{W}$ as a copy of $w$ such that $\tilde{w}$ is independent from $D_p$ by marginalizing over $D_p$, i.e., $\tilde{w} \sim \mathbb{E}_{D_p} P(w|D_p)$. We define for any $\lambda \in \mathbb{R}$:

$$\psi_{\tilde{w},\mu}(\lambda) = \log \mathbb{E}_{\tilde{w},\mu}\left[e^{\lambda\left(R_D(\tilde{w}) - \mathbb{E}_{\tilde{w},D_p}[R_{D_p}(\tilde{w})]\right)}\right] \tag{36}$$

$$= \log \mathbb{E}_{\tilde{w},\mu}\left[e^{\lambda R_D(\tilde{w})}\right] - \lambda \mathbb{E}_{\tilde{w},D_p}[R_{D_p}(\tilde{w})]. \tag{37}$$

Using Donsker-Varadhan representation of KL divergence (Boucheron et al., 2013), we have that:

$$I(w, D_p) + KL(\tilde{\mu}\|\mu) \geq \sup_g \left\{\mathbb{E}_{w,D_p}[g(w, D)] - \log \mathbb{E}_{\tilde{w},\mu}\left[e^{g(\tilde{w},D)}\right]\right\} \tag{38}$$

$$= \lambda \mathbb{E}_{w,D_p}[R_D(w)] - \log \mathbb{E}_{\tilde{w},\mu}[e^{\lambda R_D(\tilde{w})}], \quad \forall \lambda \in \mathbb{R} \tag{39}$$

$$= \lambda \mathbb{E}_{w,D_p}[R_D(w)] - \lambda \mathbb{E}_{\tilde{w},D_p}[R_{D_p}(\tilde{w})] - \psi_{\tilde{w},\mu}(\lambda). \tag{40}$$

Since $D$ and $D_p$ are independent, we can have the following equation:

$$\lambda\left(\mathbb{E}_{w,D_p}[R_D(w)] - \mathbb{E}_{w,D_p}[R_{D_p}(w)]\right) \leq I(w, D_p) + KL(\tilde{\mu}\|\mu) + \psi_{\tilde{w},\mu}(\lambda), \quad \forall \lambda \in \mathbb{R} \tag{41}$$

$$\left(\mathbb{E}_{w,D_p}[R_D(w)] - \mathbb{E}_{w,D_p}[R_{D_p}(w)]\right) \leq \frac{1}{\lambda}[I(w, D_p) + KL(\tilde{\mu}\|\mu) + \psi_{\tilde{w},\mu}(\lambda)], \quad \forall \lambda > 0 \tag{42}$$

By the assumption that $\ell(\cdot, \cdot)$ follows a $\sigma$-subgaussian, we can have that $R_D(\tilde{w})$ follows $\frac{\sigma}{\sqrt{|D|}}$-subgaussian. Since $\psi_{\tilde{w},\mu}(\lambda)$ is the cummulant generative function of random variable $R_D(\tilde{w}) - \mathbb{E}_{\tilde{w},D_p}[R_{D_p}(\tilde{w})]$, which is also $\frac{\sigma}{\sqrt{|D|}}$-subgaussian due to linearity, we have that:

$$\psi_{\tilde{w},\mu}(\lambda) \leq \frac{\lambda^2 \sigma^2}{2|D|}, \quad \forall \lambda \in \mathbb{R}. \tag{43}$$

Similar analysis as in the proof of Lemma B.1, we can derive that:

$$\left|\mathbb{E}_{w,D_p}\left[R_D(w) - R_{D_p}(w)\right]\right| \leq 2\sqrt{\frac{\sigma^2[I(w, D_p) + KL(\tilde{\mu}\|\mu)]}{2|D|}}, \tag{44}$$

which concludes the proof. □

### B.3. Proof of Corollary 5.3

**Corollary B.3.** *The utility bound and the generalization of $\mathcal{M}$ depends on the generalization of learning mechanism $\mathcal{A}$, and scale at a rate of $\mathcal{O}\left(\sqrt{\sum_{k=1}^K I(\tilde{\theta}_{S,k}, \tilde{\theta}_{T,k}; D_p)}\right)$ since $I(w; D_p) \leq \sum_{k=1}^K I(\tilde{\theta}_{S,k}, \tilde{\theta}_{T,k}; D_p)$.*

*Proof.* Denote $\Phi = \{(\tilde{\theta}_{S,k}, \tilde{\theta}_{T,k})\}_{k=1}^K$. Based on the Bayesian Network in Figure 3 and the Data-processing inequality (Cover, 1999), it follows that $I(w, D_p) \leq I(\Phi, D_p)$. Based on the chain rule of mutual information (Khinchin, 2013), we can derive that:

$$I(w, D_p) \leq I(\Phi, D_p) = \sum_{k=1}^K I(\tilde{\theta}_{S,k}, \tilde{\theta}_{T,k}; D_p|\tilde{\theta}_{S,<k}, \tilde{\theta}_{T,<k}). \tag{45}$$

Since each shadow model is trained separately from the others, we can have:

$$I(w, D_p) \leq I(\Phi, D_p) = \sum_{k=1}^K I(\tilde{\theta}_{S,k}, \tilde{\theta}_{T,k}; D_p|\tilde{\theta}_{S,<k}, \tilde{\theta}_{T,<k}) = \sum_{k=1}^K I(\tilde{\theta}_{S,k}, \tilde{\theta}_{T,k}; D_p), \tag{46}$$

which concludes the proof. □

*Table 6.* Fine-tuning hyperparameters and settings used for GRAD-TRANSFORMER and all baselines.

| Hyperparameter | Value |
|---|---|
| Optimizer | AdamW |
| Learning rate | $1 \times 10^{-5}$ |
| Learning rate scheduler | Linear |
| Batch size | 1 |
| Gradient accumulation steps | 16 |
| Gradient clipping norm | 0.1 |
| L2 regularization | 0.01 |
| LoRA rank | 2 |
| Training steps | 20000 |
| Max input length | 4096 |
| Max new tokens | 512 |

# C. Experiment Details

## C.1. Training TinyLMs on clients settings

In the training process of clients' TinyLMs, we use learning rate 1e-5 with linear scheduler. We set batch size to 1 with 16 gradient accumulation steps, gradient clipping with norm 0.1, L2 regularization with parameter decay 0.01. We fine-tune query and key parameters of all attention layers of the TinyLM and the LLM with LoRA rank 2 (Hu et al., 2022). We use the AdamW optimizer (Loshchilov & Hutter, 2019). For each training process, we train the models for 20000 steps.

## C.2. Baseline descriptions and settings

We use SoTA Weak-to-Strong Knowledge Distillation (Burns et al., 2024) methods as our baselines. Although these methods do not offer data privacy for clients since they need data sharing between the TinyLM and the LLM, we use them as baselines since they are most applicable to our setting. Previous works in Data-Free Knowledge Distillation are not able to be applied in our experiments since they are only applicable to text classification tasks and do not work for more complex text generation tasks in our experiments. Below are the descriptions of baselines used:

- **W2S** (Burns et al., 2024). This baseline uses a fine-tuned weak model to provide predicted labels for the dataset. Afterwards, the strong model is fine-tuned on the dataset using these labels instead of ground truth labels. We follow their original setting to split the original dataset in half, using one half to fine-tune the weak model first, and use the other half for predicting labels using weak models and fine-tuning the large model.

- **Conf** (Burns et al., 2024). This baseline also a fine-tuned weak model to provide predicted labels for the dataset. Similarly, the strong model is fine-tuned on the dataset using these labels instead of ground truth labels. The dataset splitting is the same compared to W2S. However, this baseline uses an auxiliary confidence loss. The intuition is that in W2S, the strong model may also learn to imitate the errors of the supervisor, so we provide additional regularization towards what the strong pretrained model already internally knows to avoid this. The auxiliary confidence loss interpolates between weak-label supervision and hardened strong-model predictions:

$$\mathcal{L}_{\text{conf}}(f) = (1 - \alpha) \, \text{CE}\big(f(x), f_w(x)\big) + \alpha \, \text{CE}\big(f(x), \hat{f}_t(x)\big), \tag{47}$$

where $\text{CE}(\cdot, \cdot)$ denotes cross-entropy, $f_w(x)$ is the weak predictive distribution, and $f(x)$ is the strong model output. The hardened target is defined as $\hat{f}_t(x) = \mathbb{I}[f(x) > t]$ with an adaptive threshold $t$ chosen so that exactly half of each batch satisfies $f(x) > t$. In our experiments, we set $\alpha_{\max} = 0.5$, and linearly warm up $\alpha$ from 0 to $\alpha_{\max}$ over the first 20% of training.

- **VisSup** (Guo et al., 2024). VisSup quantify confidence through the agreement between the model's soft prediction and its corresponding hard label, where stronger alignment implies higher confidence. Based on this principle, they introduce an adaptive confidence loss that dynamically balances weak supervision and self-supervision on a per-sample basis:

$$\mathcal{L}_{\text{AC}}(f) = \big(1 - \beta(x)\big) \, \text{CE}\big(f(x), f_w(x)\big) + \beta(x) \, \text{CE}\big(f(x), \hat{f}(x)\big), \tag{48}$$

---

**Algorithm 1** Differentially Private SGD (Abadi et al., 2016)

---

**Require:** Examples $\{x_1, \ldots, x_N\}$, loss function $\mathcal{L}(\theta) = \frac{1}{N} \sum_i \mathcal{L}(\theta, x_i)$
**Require:** Parameters: learning rate $\eta_t$, noise scale $\sigma$, group size $L$, gradient norm bound $C$
1: Initialize $\theta_0$ randomly
2: **for** $t \in [T]$ **do**
3:     Take a random sample $L_t$ with sampling probability $L/N$
4:     **Compute gradient**
5:     **for** each $i \in L_t$ **do**
6:         $g_t(x_i) \leftarrow \nabla_\theta \mathcal{L}(\theta_t, x_i)$
7:     **end for**
8:     **Clip gradient**
9:     $\bar{g}_t(x_i) \leftarrow g_t(x_i) / \max\left(1, \frac{\|g_t(x_i)\|_2}{C}\right)$
10:    **Add noise**
11:    $\tilde{g}_t \leftarrow \frac{1}{L} \sum_i \bar{g}_t(x_i) + \mathcal{N}(0, \sigma^2 C^2 I)$
12:    **Descent**
13:    $\theta_{t+1} \leftarrow \theta_t - \eta_t \tilde{g}_t$
14: **end for**
15: **Output** $\theta_T$ and compute the overall privacy cost $(\varepsilon, \delta)$ using a privacy accounting method

---

with the confidence weight defined as

$$\beta(x) = \frac{\exp\big(\mathrm{CE}(f(x), \hat{f}(x))\big)}{\exp\big(\mathrm{CE}(f(x), \hat{f}(x))\big) + \exp\big(\mathrm{CE}(f(x), \hat{f}_w(x))\big)}. \tag{49}$$

Here, $\beta(x)$ is an input-dependent confidence weight, $\hat{f}_w(x)$ denotes the hard label from the weak model, and $\mathrm{CE}(\cdot, \cdot)$ is the cross-entropy loss. The first term encourages learning from weak supervision, while the second term emphasizes self-supervision from the strong model in high-confidence cases. In our experiment, we set the temperature of the loss to 2.0.

Other hyperparameters are kept the same for all baselines and GRAD-TRANSFORMER and are shown in Table 6.

## D. Differential Privacy Protection

### D.1. Preliminaries

In this experiment, we adopt Differentially Private SGD (DP-SGD) (Abadi et al., 2016) as the client-side training mechanism $\mathcal{A}$ to obtain differential privacy protection for client's unshareable data (Dwork, 2006). Firstly, we provide some basic background on differential privacy (Dwork, 2006) and the DP-SGD mechanism (Abadi et al., 2016).

**Differential Privacy.** Differential privacy (DP) (Dwork, 2006) is a formal framework that provides strong, provable privacy guarantees for algorithms operating on sensitive datasets by ensuring that their outputs are nearly indistinguishable when a single data record is modified. A randomized mechanism satisfies $(\varepsilon, \delta)$-differential privacy if the probability of any output changes by at most a factor of $e^\varepsilon$, up to a small failure probability $\delta$, between any two adjacent datasets. Differential privacy as formally defined as follows:

**Definition D.1.** (Dwork, 2006) A randomized mechanism $\mathcal{M} : \mathcal{D} \to \mathcal{R}$ with domain $\mathcal{D}$ and range $\mathcal{R}$ satisfies $(\varepsilon, \delta)$-differential privacy if for any two adjacent inputs $d, d' \in \mathcal{D}$ and for any subset of outputs $S \subseteq \mathcal{R}$, it holds that

$$\Pr[\mathcal{M}(d) \in S] \leq e^\varepsilon \Pr[\mathcal{M}(d') \in S] + \delta.$$

Differential privacy is robust to auxiliary information and supports principled composition, making it well suited for complex machine learning systems.

**Differentially Private SGD (DP-SGD).** DP-SGD (Abadi et al., 2016) is a privacy-preserving variant of stochastic gradient descent that enforces differential privacy during model training. At each iteration, gradients are computed per example, clipped to a fixed norm to bound sensitivity, averaged, and perturbed with calibrated Gaussian noise before updating model

parameters. The cumulative privacy loss over training steps is tracked using a privacy accountant, yielding an overall $(\varepsilon, \delta)$-differential privacy guarantee for the trained model. The algorithm for DP-SGD is described in Algorithm 1.

### D.2. Results

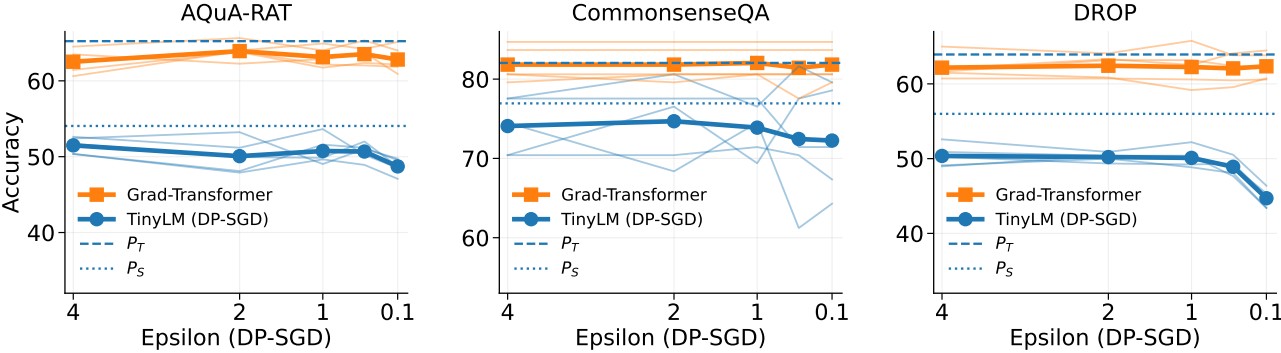

*Figure 7.* Averaged performance of GRAD-TRANSFORMER in the 5-client setting with while training client model with DP-SGD on the AQuA-RAT, CommonsenseQA, DROP. The light-colored lines show performance in each client.

We provide the results of GRAD-TRANSFORMER with DP-SGD as client's fine-tuning mechanism $\mathcal{A}$ in the 5 client setting on AQuA-RAT, Commonsense and DROP datasets in Figure 7, which is an expansion of Figure 4. This figure demonstrates that GRAD-TRANSFORMER consistently perform well using under DP-SGD fine-tuning of the TinyLMs for AQuA-RAT, CommonsenseQA and DROP datasets.

## E. Additional Experiments and Analysis

### E.1. Clients with different tasks

#### E.1.1. EACH CLIENT HAVING AN INDEPENDENT TASK

*Table 7.* Results of 3 client setting with different tasks on each client.

|  | **Metric** | **AQuA-RAT** | **CommonsenseQA** | **DROP** |
|---|---|---|---|---|
| $P_S$ | Acc | 48.43 | 77.40 | 49.36 |
| $P_T$ | Acc | 59.06 | 81.90 | 55.01 |
| **Ours** | Acc | 57.48 | 80.51 | 55.64 |
|  | PGR | 80.89 | 50.00 | 111.15 |

In this experiment, clients have different datasets, each associated with a unique task (i.e., AQuA-RAT, CommonsenseQA, or DROP). Table 7 (Appx. E) shows that GRAD-TRANSFORMER is able to approach ceiling performance (80.68% of PGR on average) and outperform the TinyLM for each client even when they have different tasks. In fact, the generated update vectors for the LLM enable the LLM to achieve 57.48, 80.51, 55.64 compared with 48.43, 77.40, 49.36 of the TinyLMs on AQuA-RAT, CommonsenseQA, and DROP correspondingly. This promising results demonstrate the ability to generalize GRAD-TRANSFORMER across clients and tasks.

#### E.1.2. EACH CLIENT HAVING A MIXTURE OF TASKS

To empirically validate robustness under realistic distribution mismatches, we construct a 3-client federated scenario where each client's data is a mixture of three distinct tasks: AQuA-RAT (math reasoning), CommonsenseQA (commonsense reasoning), and DROP (discrete reasoning). First, the public data is constructed by taking 50% of the original training set in AQuA-RAT, 80% of the original training set in CommonsenseQA, 80% of the original training set in DROP, resulting in a distribution of (0.4172, 0.0656, 0.5215) for these datasets. The private data for clients are sampled via a Dirichlet distribution with $\alpha = [1, 1, 1]$ from the remaining data samples, each client having 2000 data samples in total. This creates substantial heterogeneity across clients, and importantly, between each client's distribution and the shadow data distribution used to train GRAD-TRANSFORMER. Note that in this experiment, we train the TinyLM for each client, and feed the update vectors

to the GRAD-TRANSFORMER individually to get the updated LLM (Ours) instead of aggregating fine-tuned TinyLMs from all clients. The data of each client is split into training set and test set with ratio 90:10. The resulting data distributions are shown in Table 8. Note the substantial distribution shifts: Client 2 is heavily skewed toward DROP (79.86%) with minimal CommonsenseQA (4.75%), while Client 3 is dominated by CommonsenseQA (67.71%) with almost no DROP (1.71%). Additionally, the private data distribution differs from all three clients.

*Table 8.* The distribution of data from different tasks in different clients.

|  | AQuA-RAT | CommonsenseQA | DROP |
|---|---|---|---|
| Client 1 | 0.3374 | 0.3278 | 0.3346 |
| Client 2 | 0.1538 | 0.0475 | 0.7986 |
| Client 3 | 0.3056 | 0.6771 | 0.0171 |
| Public data | 0.4172 | 0.0656 | 0.5215 |

*Table 9.* Accuracy of TinyLM, LLM and Ours for each client.

|  | Client 1 | Client 2 | Client 3 |
|---|---|---|---|
| $P_S$ (TinyLM) | 54 | 46.5 | 63.5 |
| Ours | 68 | 49.5 | 69 |
| $P_T$ (LLM) | 66 | 55 | 70.5 |

Despite the substantial distributional mismatch, GRAD-TRANSFORMER consistently improves over the TinyLM baseline for all clients, by 25.9%, 6.5%, and 8.7% respectively. Client 1 (balanced distribution, moderate divergence from shadow data) even exceeds the LLM ceiling $P_T$, while Client 2 (highest divergence from shadow data) shows the smallest but still positive improvement, consistent with Theorem 5.2 that utility degrades gracefully with $\mathrm{KL}(\tilde{\mu}\|\mu)$ rather than collapsing.

These results suggest that while perfect distributional alignment is ideal, GRAD-TRANSFORMER does not require it to deliver meaningful gains. The framework captures sufficiently general correlations between TinyLM and LLM update vectors during training on $D_p$ to remain effective under realistic distribution shifts.

### E.2. Number of update vector tuples.

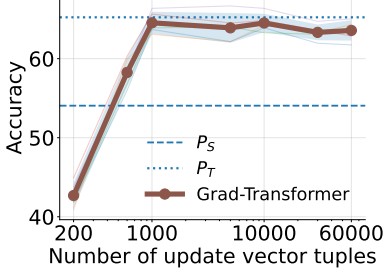

*Figure 8.* Ablation study on the number of update vector tuples and Grad-Transformer's performance on AQuA-RAT in the 5 client setting (log-scaled). The red line shows average performance across 5 clients. The light-colored lines show performance of each client using GRAD-TRANSFORMER.

Figure 8 shows the performance of GRAD-TRANSFORMER on AQuA-RAT when using different numbers of update vector tuples for training. We observe that at 1,000 update vector tuples, GRAD-TRANSFORMER has converged to its optimal performance. This observation directly shows the minimal computing costs needed for GRAD-TRANSFORMER to reach its optimal performance.

### E.3. Time consumption

In this section, we provide the time consumption analysis for GRAD-TRANSFORMER framework. To ensure fair comparison, all time consumption analysis are run using $1\times$ NVIDIA A100 80GB GPU.

*Table 10.* Time saved (minutes) for each client when fine-tuning 3B-scale TinyLLM compared to fine-tuning 7B-scale LLM, following the main experiments in Table 1 and 3. AQR: AQuA-RAT.

| Stage | AQR | GSM8K | DROP |
|---|---|---|---|
| Fine-tune 3B-TinyLM (1 client) | 53.70 | 50.47 | 101.13 |
| Fine-tune 7B-LLM (1 client) | 71.54 | 56.45 | 160.43 |
| Time saved using Grad-Transformer | 17.84 | 5.98 | 59.30 |
| Time reduction in percentage | 24.93% | 10.59% | 36.96% |

*Table 11.* GRAD-TRANSFORMER framework time consumption analysis using one NVIDIA A100 80GB GPU. Time consumption is computed with Qwen2.5-3B-Instruct as TinyLM and Qwen2.5-7B-Instruct as LLM as in the main experiments of the paper. Time unit is minutes.

*(a)* Update vectors curation cost

| Stage | AQuA-RAT | GSM8K | DROP |
|---|---|---|---|
| Train a shadow TinyLM | 7.15 | 5.98 | 18.67 |
| Train a shadow LLM | 7.03 | 3.76 | 13.17 |
| Train a pair of shadow models | 14.18 | 9.74 | 31.84 |

*(b)* GRAD-TRANSFORMER total training cost

| Stage | AQuA-RAT | GSM8K | DROP |
|---|---|---|---|
| Update vectors curation | 70.92 | 48.70 | 159.20 |
| Train GRAD-TRANSFORMER | 477.00 | 521.50 | 735.00 |
| **Total** | **484.03** | **570.20** | **894.20** |

Using GRAD-TRANSFORMER to update a 7B LLM using update vectors from a 3B TinyLM as in experiments in Table 1 and Table 3 reduces 24.16% training time on average across AQuA-RAT, GSM8K and DROP datasets, when compared to directly fine-tuning a 7B LLM model on client datasets (Table 10).

In Table 11, we show the time costs for update vectors curation and GRAD-TRANSFORMER training. In Table 8a, we show the time taken to train a pair of shadow models during update vectors curation for AQuA-RAT, GSM8K, DROP. In this process, we train the shadow TinyLM and LLM until convergence. Depending on the dataset, training a pair of shadow models takes from 9 to 32 minutes. Table 8b shows the total cost for GRAD-TRANSFORMER framework, which includes update vectors curation and training the GRAD-TRANSFORMER. The update vectors curation time shown is the time taken to curate 1,000 update vector tuples (since this is the sufficient number of update vector tuples as shown in Figure 8). The GRAD-TRANSFORMER training time shown includes time taken to train the GRAD-TRANSFORMER for 30 epochs. Depending on the dataset, it takes from 484.03 to 894.20 minutes to curate update vectors and train the GRAD-TRANSFORMER.

# F. Limitations and Future Work

While GRAD-TRANSFORMER demonstrates remarkable performance across various language modeling and reasoning tasks, there are several limitations and potential areas for further discussion. Firstly, the theoretical bound in Theorem 5.2 demonstrates that the performance of GRAD-TRANSFORMER is dependent on the KL divergence between the public data and private data. For highly sensitive, niche domains, curating an aligned public dataset is difficult, which may limit the performance of GRAD-TRANSFORMER. Secondly, we have not designed GRAD-TRANSFORMER for sparse architectures like Mixture-of-Experts (Zhou et al., 2022), where LoRA update vectors may exhibit complex routing behaviors. In future work, we extend GRAD-TRANSFORMER to cases where the distribution of private data shifts significantly from the distribution of public data, as well as redesign GRAD-TRANSFORMER for sparse architectures.

