# OpenReview forum: "Gradient Transformer: Learning to Generate Updates for LLMs"
_ICML.cc/2026/Conference — ICML 2026 regular_

### Official Review · Reviewer_KUYz · 2026-03-08

**Soundness:** 3
**Presentation:** 3
**Significance:** 2
**Originality:** 3
**Overall Recommendation:** 4
**Confidence:** 3

**Summary:**

This paper presents GRAD-TRANSFORMER, a data-free, weak-to-strong knowledge distillation framework designed to generate update vectors for Large Language Models (LLMs) based on TinyLMs fine-tuned on private data. By keeping the private data completely isolated on the client side, the method circumvents standard privacy and data-sharing bottlenecks. The core contribution is a Transformer-based encoder-decoder architecture that learns the mapping between the update vectors of TinyLMs and LLMs via block-wise parameter representations, trained entirely on shadow datasets curated from public data. The authors evaluate the method on six benchmark datasets encompassing language modeling and reasoning in both single- and multi-client scenarios, demonstrating significant performance gains over existing baselines. Theoretical generalization bounds, robustness checks under strict differential privacy, and scaling ablations further support the empirical findings.

**Compliance With Llm Reviewing Policy:**

Affirmed.

**Final Justification:**

This paper presents a data-free framework for mapping parameter updates from TinyLMs to LLMs. The authors' rebuttal addressed my initial concerns regarding empirical soundness. While extending this framework to more up-to-date or sparse architectures remains a valuable direction for future work, the current contributions are well-supported. With my prior reservations largely resolved, I have raised my score to 4 and maintain a slightly positive stance on this submission.

**Key Questions For Authors:**

See Weakness.

**Limitations:**

The authors should explicitly discuss the following two technical limitations in a dedicated section: 1) Dependence on Shadow Data Alignment: The theoretical bounds (Theorem 5.2) demonstrate a reliance on the KL divergence between the shadow and private datasets. The authors should acknowledge that for highly sensitive, niche domains, curating an aligned public shadow dataset is practically infeasible, inherently limiting the method's utility. 2) Generalization to Complex Architectures: The current block-wise mapping paradigm assumes dense, uniform parameter spaces. A significant limitation is the framework's unproven applicability to models with sparse routing (e.g., MoE), where expert-aware updating and inherent task conflicts within adapters would likely disrupt the current translation mechanism.

**Strengths And Weaknesses:**

Strength:

1.The paper tackles a critical bottleneck in federated and decentralized LLM adaptation. Bypassing the need to either share raw data or fine-tune massive models on resource-constrained client devices is a highly valuable contribution to the field.

2.Framing model adaptation as a sequence-to-sequence translation of block-wise update vectors is an elegant solution. This design efficiently handles the massive dimensionality of LLM parameters, making the mapping process scalable.

3.The evaluation is thorough. Providing theoretical utility bounds in an information-theoretic framework (Theorem 5.2), alongside practical ablations on differential privacy (DP-SGD) and multi-client scaling, bridges the gap between deep learning theory and deployment realities.



Weakness:

1.The block-wise partitioning (Figure 2) is intuitively justified for scalability, but it remains an arbitrary design choice. There is no empirical ablation comparing this against alternative chunking strategies, layer-wise mappings, or higher-level semantic parameter representations to justify why block-wise is optimal.

2.The paper lacks a granular, task-specific error analysis. It is currently unclear what specific types of reasoning errors or knowledge gaps are successfully bridged by the Grad-Transformer updates, and which types consistently fail to map from the TinyLM to the LLM.

3.While Theorem 5.2 indicates that performance scales with data alignment, the real-world heterogeneity of private versus shadow datasets is underexplored. In practical federated settings, client data is heavily non-IID and often strictly out-of-distribution (OOD) relative to public data. The empirical robustness under severe distribution shifts needs validation.

4.The evaluation is strictly confined to dense models within the same family (Qwen2.5). The framework does not address how the Grad-Transformer handles structural mismatches, or how it would scale to sparse architectures like Mixture-of-Experts (MoE), where LoRA updates exhibit complex routing behaviors and task conflicts that resist simple linear block-wise alignment.

---

> ### Author Rebuttal · Authors · 2026-03-31
>
> Thank you for your valuable feedback. Your positive remarks about our "highly valuable contribution," "elegant solution," "thorough evaluation" is highly motivating. We address each concern below.
>
> **W1.** We select block-wise partitioning for three reasons: (1) concatenating all weights into a single vector is infeasible (Section 3, Challenge 1), (2) grouping weights within each attention block preserves the functional unit where these weights are jointly optimized and semantically coupled, and (3) sequential block ordering aligns with autoregressive generation, allowing the decoder to condition each block on previously generated blocks.
>
> **Table 1: Ablation on block-wise segmentation (AQuA-RAT, 5 clients).**
>
> | | Client 1 | Client 2 | Client 3 | Client 4 | Client 5 | Avg. |
> |---|---|---|---|---|---|---|
> | $P_S$ | 53.95 | 54.26 | 53.64 | 54.15 | 54.26 | 54.05 |
> | Ours | 61.74 | 64.31 | 63.90 | 62.87 | 65.03 | 63.57 |
> | Q-V separately | 40.51 | 40.72 | 41.85 | 43.79 | 43.38 | 42.05 |
> | Reverse block order | 38.15 | 38.05 | 40.21 | 42.05 | 43.08 | 40.31 |
> | $P_T$ | 64.92 | 66.05 | 64.51 | 64.21 | 66.46 | 65.23 |
>
> **Q-V separately** trains two $\texttt{Grad-Transformer}$s on query and value updates independently. In the original method, we concatenate query and value weight updates of each block to train a single $\texttt{Grad-Transformer}$. This separation breaks the intra-block correlation between query and value weight updates, causing performance to drop sharply to 42.05%. **Reverse block order** reverses the block ordering of weight update vectors before feeding them to $\texttt{Grad-Transformer}$, disrupting the natural sequential dependency between layers. Performance drops further to 40.31%, confirming that the autoregressive decoder relies on correct layer ordering to capture inter-block correlations. These results demonstrate that our block-wise strategy, preserving both intra-block weight correlations and inter-block sequential structure, is critical.
>
> **W2 & W3.** To jointly address both concerns, we designed a new experiment that reveals task-specific performance patterns and validates robustness under non-IID distribution shifts. We construct a 3-client scenario where each client's data is a Dirichlet-sampled ($\alpha=[1,1,1]$) mixture of AQuA-RAT, CommonsenseQA, and DROP (2000 samples each). We use the Grad-Transformer on each client's TinyLM individually instead of aggregating. The public data distribution is (0.42, 0.07, 0.52), while clients differ substantially as in Table 2:
>
> **Table 2: Data distributions across clients.**
>
> | | AQuA-RAT | CommonsenseQA | DROP |
> |---|---|---|---|
> | Client 1 | 0.3374 | 0.3278 | 0.3346 |
> | Client 2 | 0.1538 | 0.0475 | 0.7986 |
> | Client 3 | 0.3056 | 0.6771 | 0.0171 |
> | Shadow | 0.4172 | 0.0656 | 0.5215 |
>
> **Table 3: Accuracy results.**
>
> | | Client 1 | Client 2 | Client 3 |
> |---|---|---|---|
> | $P_S$ (TinyLM) | 54 | 46.5 | 63.5 |
> | Ours | 68 | 49.5 | 69 |
> | $P_T$ (LLM) | 66 | 55 | 70.5 |
>
> As in Table 3, client 1 (balanced) achieves the strongest improvement (+14%), suggesting balanced mixtures enable effective cross-task transfer. Client 3 (CommonsenseQA-heavy) shows solid gains (+5.5%), indicating commonsense reasoning transfers well through update vectors. Client 2 (DROP-heavy) shows smaller improvement (+3%) and a larger gap to $P_T$, suggesting discrete reasoning compositions are more challenging. A promising direction is applying RL [1] as the learning mechanism $\mathcal{A}$ on the client side to produce higher-quality TinyLM update vectors, and incorporating RL-based post-processing [2] on the generated LLM updates. We leave this to future work.
>
> Despite substantial distribution shifts between each client and the public dataset, $\texttt{Grad-Transformer}$ improves over TinyLM for all clients (by 25.9%, 6.5%, and 8.7%). This validates our meaningful performance under realistic non-IID heterogeneity between clients and distribution shift of client data relative to public data, supporting its practical applicability in real-world settings.
>
> **W4.** We acknowledge that the current pipeline is limited to conventional dense LLM architectures and is not readily applicable to more advanced designs such as MoE. However, we extend the discussion of the potential of \texttt{Grad-Transformer} to generalize to recent advanced structures, such as MoE. Since MoE replaces the FFN layer with routed expert FFN layers [3], $\texttt{Grad-Transformer}$ could potentially update the self-attention layers and expert FFN weights, while users fine-tune only the lightweight gating layers with their data. We leave this as an important direction for future work.
>
> [1] Ouyang et al., Training Language Models to Follow Instructions with Human Feedback. NeurIPS 2022.
>
> [2] Guo et al., DeepSeek-R1: Incentivizing Reasoning Capability in LLMs via Reinforcement Learning. arXiv 2025.
>
> [3] Zhou et al., Mixture-of-Experts with Expert Choice Routing. NeurIPS 2022.

---

> > ### Author Rebuttal · Reviewer_KUYz · 2026-04-01
> >
> > I appreciate the authors' hard work and detailed responses.
> >
> > Since the majority of my concerns have been resolved, I have updated my score to 4. Moreover, the paper's contribution would be even more compelling if the authors could adapt the current framework to more up-to-date models.

---

### Official Review · Reviewer_F2Ug · 2026-03-08

**Soundness:** 2
**Presentation:** 3
**Significance:** 2
**Originality:** 2
**Overall Recommendation:** 4
**Confidence:** 4

**Summary:**

This paper focuses on the challenge that how to perform weak-to-strong knowledge distillation when the training data is private data. The proposed data-free distillation method uses a vector of parameter changes rather than logits.

**Compliance With Llm Reviewing Policy:**

Affirmed.

**Key Questions For Authors:**

1 The size and quality of the public dataset including how similar it is to the downstream task have a strong influencing on the final distillation results. Could you please provide a detailed experimental analysis about the public dataset?

2 In Table 1 and Table 2, on some tasks, the performance of ours even outperform than the PT (ceiling performance), could you please explain the phenomenon?

**Limitations:**

The construction of the public dataset and the overly simple handing of multiple clients may limit its application in real-world.

**Strengths And Weaknesses:**

Strengths

1 This paper is well-structured and logically organized。The method is described clearly.

2 In the experimental section, experiments are conducted across multiple scales of teacher and student models, demonstrating that the method has good robustness.

Weaknesses

1 As shown in equation 5.2, the KL-divergence between the distribution of the public dataset and that of the downstream task has a strong influence on  the utility bound. However, since the private data is unavailable, finding a suitable public dataset may not be easy in the real-world applications.

2 the block-wise segment of the models may rigidly break the coordination and connections across different layers and may cut off the truly important global information.

3 In Section 4.3, simply averaging the parameter updates from N clients may lead to the loss of key signals or introduce noise, conflicts, especially when the data distributions across different clients differ largely.

---

> ### Author Rebuttal · Authors · 2026-03-31
>
> Thank you for your thoughtful comments. We deeply appreciate your recognition of our work as "well-structured," "logically organized", and having "good robustness." We address each concern below.
>
> **W1.** We acknowledge that finding a public dataset $D_p$ whose distribution $\tilde{\mu}$ closely aligns with the unknown private distribution $\mu$ is challenging, and $\mathrm{KL}(\tilde{\mu} \| \mu)$ in Theorem 5.2 directly governs the utility bound. However, we emphasize two practical mitigations and provide new empirical evidence.
>
> **Practical mitigations.** The service provider typically knows the downstream task $\tau$ (e.g., math reasoning), which significantly narrows the candidate public datasets. When task-specific data is scarce, $D_p$ can cover broad general knowledge to capture correlations between TinyLM and LLM update vectors even without precise distributional alignment.
>
> **Robustness under distribution shift.** We construct a 3-client scenario where each client's data is a Dirichlet-sampled ($\alpha=[1,1,1]$) mixture of AQuA-RAT, CommonsenseQA, and DROP (2000 samples each client). We split each client's data to training set and test set with ratio 90:10. We use the Grad-Transformer on each client's TinyLM individually instead of aggregating. The public data distribution is (0.42, 0.07, 0.52), while clients differ substantially:
>
> **Table 1: Data distributions across clients.**
> | | AQuA-RAT | CommonsenseQA | DROP |
> |---|---|---|---|
> | Client 1 | 0.3374 | 0.3278 | 0.3346 |
> | Client 2 | 0.1538 | 0.0475 | 0.7986 |
> | Client 3 | 0.3056 | 0.6771 | 0.0171 |
> | Public | 0.4172 | 0.0656 | 0.5215 |
>
> **Table 2: Accuracy results.**
> | | Client 1 | Client 2 | Client 3 |
> |---|---|---|---|
> | $P_S$ (TinyLM) | 54 | 46.5 | 63.5 |
> | Ours | 68 | 49.5 | 69 |
> | $P_T$ (LLM) | 66 | 55 | 70.5 |
>
> As in Table 2, despite substantial distributional mismatch between each client and public data, $\texttt{Grad-Transformer}$ improves over TinyLM for all clients by 25.9%, 6.5%, and 8.7% respectively. These results suggest that while distributional alignment is ideal, Grad-Transformer does not require it to deliver meaningful gains. We will add practical guidelines for public dataset selection in the revised version.
>
> **W2.** Our architecture preserves global coordination through two mechanisms. First, the encoder processes the entire sequence of TinyLM block-wise update vectors jointly: self-attention captures dependencies and correlations across all blocks before any decoding occurs. Second, the decoder generates LLM updates autoregressively, conditioning each block on both the full encoder output and all previously generated blocks. This explicitly models inter-layer dependencies. Block-wise segmentation serves as a tokenization strategy analogous to how Transformers segment text into tokens without losing document-level coherence [1]. Our strong empirical results provide evidence that the architecture effectively captures cross-layer coordination despite the block-wise decomposition.
>
> **W3.** Averaging model updates is well-established in FL through FedAvg [2], which converges under non-IID conditions [3]. Unlike standard FL where gradients are averaged at intermediate steps, we aggregate update vectors after each client's TinyLM has already converged. Table 6 (Appendix E.1) shows that with 3 clients on entirely different tasks, $\texttt{Grad-Transformer}$ achieves 80.68% average PGR, substantially outperforming each client's TinyLM. The Dirichlet experiment from W1 further confirms robustness under substantial distributional heterogeneity across clients.
>
> **Q1.** We analyze three dimensions. **Size:** Figure 7 shows performance increases monotonically with shadow dataset size (57.87% at 256 samples to 61.02% at 1024), consistent with Lemma 5.1 predicting that larger $D_p$ reduces $I(w; D_p)$ and improves generalization. **Tuples:** Figure 8 shows convergence at ~1000 update vector tuples, demonstrating modest curation costs. **Distribution:** The Dirichlet experiment in W1 shows consistent gains despite no client matching the public data distribution. **Guidance:** Practitioners should prioritize (1) sufficient dataset size ($\geq$1024 samples per shadow dataset), (2) ~1000 update vector tuples, and (3) reasonable distributional alignment with the downstream task.
>
> **Q2.** $\texttt{Grad-Transformer}$ is trained on 60,000 update vector tuples from 300 shadow datasets, exposing it to diverse fine-tuning trajectories, a form of implicit ensembling. This produces update vectors that capture robust transformation patterns, avoiding the bias and steep local minima that single-trajectory fine-tuning ($P_T$) on a private dataset may encounter.
>
> [1] Raffel et al., Exploring the Limits of Transfer Learning with a Unified Text-to-Text Transformer. JMLR 2020.
>
> [2] McMahan et al., Communication-Efficient Learning of Deep Networks from Decentralized Data. AISTATS 2017.
>
> [3] Li et al., On the Convergence of FedAvg on Non-IID Data. ICLR 2020.

---

> > ### Author Rebuttal · Reviewer_F2Ug · 2026-04-05
> >
> > Thanks for you response. I will keep my positive score.

---

### Official Review · Reviewer_J1bh · 2026-03-12

**Soundness:** 2
**Presentation:** 3
**Significance:** 3
**Originality:** 2
**Overall Recommendation:** 4
**Confidence:** 2

**Summary:**

The paper addresses the challenge that many organizations cannot afford to fine-tune large language models on private datasets due to limited computational resources, while fine-tuning only tinyLMs usually yields inferior performance. To address this issue, the paper proposes a data-free weak-to-strong knowledge distillation framework, where the update vectors of an LLM are generated by transforming the fine-tuning updates obtained from tinyLMs. Specifically, the authors introduce GRAD-TRANSFORMER, which learns to map tinyLMs update vectors into the update space of LLMs, enabling efficient adaptation of larger models without directly fine-tuning them on private data. The paper positions this as the first data-free weak-to-strong distillation approach for LLM adaptation. In addition, the authors provide theoretical analysis of the proposed method and conduct extensive experiments to demonstrate its effectiveness.

**Compliance With Llm Reviewing Policy:**

Affirmed.

**Final Justification:**

Thank you for the detailed rebuttal. The responses address several of my main concerns. I think the overall rebuttal satisfactory and believe the paper is stronger. I will revise my score upward.

**Key Questions For Authors:**

See weaknesses.

**Limitations:**

There is a typo on line 243: “iw” should be corrected to “is”.

**Strengths And Weaknesses:**

**Strengths:**

- GRAD-TRANSFORMER presents an weak-to-strong knowledge distillation perspective for adapting LLMs under limited computational resources, which formulates the problem as transferring task adaptation signals from tinyLMs to LLMs via update vectors.

- GRAD-TRANSFORMER can use tinyLMs to update LLMs without direct access to private data.

- Extensive experiments across multiple reasoning and generation datasets demonstrate the effectiveness of GRAD-TRANSFORMER.

**Weaknesses:**

- The main novelty of the work appears to lie more in the problem formulation than in the algorithmic design. The proposed method essentially learns a mapping between tinyLM and LLM update vectors using a transformer architecture. While the weak-to-strong update transfer perspective is interesting, the underlying technical implementation is relatively straightforward.

- The paper uses DP-SGD for client-side privacy protection, but does not evaluate the privacy risk of the TinyLM update vectors themselves. Could these vectors leak private information, such as data attributes or even raw training samples? It would strengthen the paper to include privacy attack baselines or leakage analysis on the transmitted update vectors.

- Why are no simple update-mapping baselines, such as linear layers, MLPs, or lightweight CNNs, included for comparison? These experiments would help clarify whether the Transformer architecture is truly necessary.

- The ablation studies vary the size of the shadow datasets and the number of update vector tuples, but do not examine the effect of block-wise segmentation granularity. How sensitive is the method to the chosen block partition strategy, and would finer or coarser segmentation affect performance or efficiency?

---

> ### Author Rebuttal · Authors · 2026-03-31
>
> Thank you for your valuable feedback. We are encouraged by your praise for our weak-to-strong distillation perspective, privacy-preserving capability, and extensive experiments. We address each concern below.
>
> **W1:** We would like to take this opportunity to clarify our contributions: (1) While the encoder-decoder builds on Transformer components, the novelty lies in adapting it to the space of update vectors (millions of parameters vs. hundreds of embedding features per token). Classical approaches (MLPs, VAEs) would require trillions of parameters (Section 3). Our block-wise segmentation decomposes billion-parameter update vectors into token-like units, with custom embeddings ($W_S^{\text{emb}}$, $W_T^{\text{emb}}$) and output projection ($W_{\text{out}}$) bridging heterogeneous parameter spaces, a novel architecture for a different domain. (2) We are the first to propose data-free weak-to-strong knowledge distillation through direct update vector transformation, shifting from aligning logits or synthesizing proxy data to learning structural correlations in update vector space across model scales. This opens a new direction for cost-effective privacy-preserving LLM adaptation. (3) We provide rigorous theoretical analysis with generalization guarantees grounded in information theory, requiring careful Bayesian network construction and novel application of the data processing inequality.
>
> **W2:** We comprehensively address this from multiple angles, as follows. (1) Existing attacks are ineffective in $\texttt{Grad-Transformer}$: Membership inference attacks perform near random guessing on LLM predictions [1]. Reconstruction and inversion attacks in the current literature focus on inverting prompts [2] or reconstructing text from token embeddings [3], neither of which applies to update vectors in our setting. (2) $\texttt{Grad-Transformer}$ transmits update vectors, which provide stronger privacy protection compared with full model gradients, as shown in GradientHide [4], aggregated gradients over multiple steps better resist gradient inversion attacks. (3) Our DP-SGD experiments (Figures 5–6) maintain strong performance under strict differential privacy ($\varepsilon=0.1$), providing provable protection against any attack on transmitted vectors, including MIAs and RAs. We will add a dedicated privacy discussion to the revised paper.
>
> **W3:** 1) A linear layer/MLP would need billions to trillions of parameters. In our setting (Qwen2.5-3B $\to$ Qwen2.5-7B), a single linear layer with hidden size 4096 requires $4.47 \times 10^9$ parameters. Even if feasible, these architectures treat all blocks independently and cannot capture sequential dependencies between layers essential for coherent update generation. 2) A CNN lacks the global attention mechanism needed to condition each output block on the full set of encoder representations simultaneously. 3) Generating update vectors is fundamentally a sequence-to-sequence task: mapping $L_S$ input blocks to $L_T$ output blocks with autoregressive dependencies where each $\hat{\delta}_{T,k}^{j}$ is conditioned on all encoder states and all previously generated blocks (Eq. 8, 11), making the Transformer the natural architecture.
>
> **W4:** We select block-wise segmentation because: (1) concatenating all weights into a single vector is infeasible (Section 3), (2) attention blocks are natural functional units of the Transformer, and (3) sequential block ordering aligns with autoregressive generation. We conduct ablations with two alternative strategies on AQuA-RAT (5-client):
>
> **Table 1: Ablation on block-wise segmentation (AQuA-RAT, 5 clients).**
>
> | | Client 1 | Client 2 | Client 3 | Client 4 | Client 5 | Avg. |
> |---|---|---|---|---|---|---|
> | $P_S$ | 53.95 | 54.26 | 53.64 | 54.15 | 54.26 | 54.05 |
> | Ours | 61.74 | 64.31 | 63.90 | 62.87 | 65.03 | 63.57 |
> | Q-V separately | 40.51 | 40.72 | 41.85 | 43.79 | 43.38 | 42.05 |
> | Reverse block order | 38.15 | 38.05 | 40.21 | 42.05 | 43.08 | 40.31 |
> | $P_T$ | 64.92 | 66.05 | 64.51 | 64.21 | 66.46 | 65.23 |
>
> **Q-V separately** trains two $\texttt{Grad-Transformer}$s on query and value updates independently, breaking intra-block correlation: performance drops sharply to 42.05%. **Reverse block order** disrupts the natural layer ordering that the autoregressive decoder relies on: performance drops further to 40.31%. These results demonstrate that our segmentation strategy, preserving both intra-block weight correlations and inter-block sequential structure, is critical to $\texttt{Grad-Transformer}$'s effectiveness.
>
> [1] Duan et al., Do Membership Inference Attacks Work on Large Language Models? COLM 2024.
>
> [2] Nazir et al., Better Language Model Inversion by Compactly Representing Next-Token Distributions. NeurIPS 2025.
>
> [3] Zhang et al., Universal Zero-shot Embedding Inversion. arXiv 2025.
>
> [4] Cheng et al., GradientHide: Federated Learning with Two-Stage Local Update for Defending Against Gradient Inversion Attack, 2026.

---

> > ### Author Rebuttal · Reviewer_J1bh · 2026-04-04
> >
> > Thank you for the detailed rebuttal. The responses address several of my main concerns. I think the overall rebuttal satisfactory and believe the paper is stronger. I will revise my score upward.

---

> > > ### Author Response · Authors · 2026-04-07
> > >
> > > We sincerely appreciate your careful evaluation of our responses and additional experiments throughout the review process. We are glad that the clarifications and new results addressed your concerns, and we are grateful for the constructive feedback.

---

### Official Review · Reviewer_rC2f · 2026-03-13

**Soundness:** 3
**Presentation:** 3
**Significance:** 3
**Originality:** 3
**Overall Recommendation:** 4
**Confidence:** 2

**Summary:**

This paper studies the problem of privacy-preserving LLM fine-tuning when private data cannot be shared with external service providers. To address this challenge, the authors propose Grad-Transformer, a framework that generates update vectors for large language models based on the update vectors of TinyLMs fine-tuned on private data. Instead of accessing the private data directly, the method learns a transformation between TinyLM and LLM update vectors using paired updates derived from shadow datasets. The approach is evaluated on multiple reasoning and language generation benchmarks, and the authors further provide theoretical analyses on generalization and utility, along with experiments under multi-client and differential privacy settings to demonstrate the effectiveness and robustness of the proposed method.

**Compliance With Llm Reviewing Policy:**

Affirmed.

**Final Justification:**

The authors have addressed my main concerns during the rebuttal. In particular, they clarified the role of the theoretical analysis and provided additional discussion on limitations and potential failure cases.

Thanks for author's response. I appreciate the clarifications provided, and I am satisfied with how my concerns were addressed. I will keep my score unchanged.

**Key Questions For Authors:**

See Weakness and Strengths part.

**Limitations:**

Yes

**Strengths And Weaknesses:**

**Soundness:**

**Strengths:**

1. The paper clearly formulates the learning objective as transforming TinyLM update vectors \Delta\theta_S into LLM updates \Delta\theta_T, which provides a well-defined mechanism for knowledge transfer without accessing private data.

2. The proposed Grad-Transformer decomposes update vectors into block-wise tokens corresponding to transformer layers, enabling scalable modeling of extremely high-dimensional parameter updates.

3. The paper derives generalization and utility bounds based on information-theoretic analysis, showing how the performance depends on mutual information and distribution shift between public and private datasets.

**Weakness:**

1. The generalization bounds rely on standard information-theoretic assumptions (e.g., sub-Gaussian loss) and mainly describe the effect of dataset size and mutual information. The analysis does not provide insights specific to update vector transformation.

2. Limited analysis of failure cases. The paper mainly reports positive results but provides limited discussion on scenarios where the transformation

 **Presentation:**

Overall structure is clear and well organized. And figures effectively illustrate the pipeline.

**Significance:**

Privacy-preserving LLM adaptation is a key challenge for organizations that cannot share data with service providers. And by sharing update vectors rather than data, the approach enables multi-organization model improvement.

---

> ### Author Rebuttal · Authors · 2026-03-31
>
> Thank you for your insightful feedback. We are greatly encouraged by your appreciation of the well-defined formulation of our learning objective, the scalability of our block-wise decomposition approach, and the rigor of our information-theoretic analysis.
>
> We have carefully considered each point and provide our detailed responses below.
>
> **Weakness 1**
>
> We thank the reviewer for raising this point, and here is our response. **(1) On the focus of the bound.** The primary objective of $\texttt{Grad-Transformer}$ is to update the LLM $\theta_T$ to achieve high utility on a downstream task. Accordingly, Theorem 5.2 bounds the loss of the *updated LLM on the private dataset*, which is the operationally meaningful quantity in our setting. A bound on the transformation of the update vector itself would not directly capture this end-to-end utility guarantee. **(2) On convergence.** Since $\texttt{Grad-Transformer}$ is built upon the transformer/attention architecture, its convergence analysis can be adapted from existing results [1, 2] under standard assumptions, and we can adopt these accordingly to provide convergence analysis on the update vector transformation mechanism $\mathcal{M}$. **(3) On how to construct the update vector.** The key structural insight in our analysis stems from how the update vector $\Delta\theta = \theta^* - \theta^0$ is constructed: given a fixed pretrained model $\theta^0$ and a dataset $D$, all randomness in $\Delta\theta$ is induced solely by the learning mechanism $\mathcal{A}$ used to obtain $\theta^*$. Thus, we bound the impact of the update vector on model utility via Corollary 5.3, which precisely characterizes how the choice of $\mathcal{A}$ propagates through the update vectors and affects downstream model utility. **(4) On the generalization of $\texttt{Grad-Transformer}$ to larger models.** Finally, we notice that recent works [3] focus on deriving a generalization bound on the context length of the LLMs, which we can leverage to derive a generalization bound on the capability of $\texttt{Grad-Transformer}$ generating update vectors for larger models (i.e., $>70$B models). However, due to the enormous parameter space of LLMs and the intractability of their training processes, deriving such a bound is challenging and requires significant effort. We will leave this analysis for future work and focus on this bound in the extension of this work.
>
> **Weakness 2**
>
> As we discussed in the paper (Section 6, line 436), a failure case of $\texttt{Grad-Transformer}$ is when we scale the LLM to the scale of 32B. We have added experimental results for the failure case when scaling to a 32B target LLM (Table 1).
>
> **Table 1: Scaling to 32B LLM on AQuA-RAT (5-client setting).**
>
> | | Client 1 | Client 2 | Client 3 | Client 4 | Client 5 | Avg. |
> |---|---|---|---|---|---|---|
> | $P_S$ | 53.95 | 54.26 | 53.64 | 54.15 | 54.26 | 54.05 |
> | Ours (3B $\to$ 14B) | 68.41 | 65.85 | 67.59 | 69.33 | 67.49 | 67.73 |
> | $P_T$ (14B) | 68.10 | 66.77 | 67.08 | 69.13 | 69.13 | 68.04 |
> | Ours (3B $\to$ 32B) | 55.59 | 51.08 | 52.21 | 56.92 | 54.26 | 54.01 |
> | $P_T$ (32B) | 75.28 | 74.77 | 75.18 | 77.44 | 78.77 | 76.29 |
>
> When using a 3B TinyLM to generate updates for a 32B LLM, performance drops to 54.01% (vs. 76.29% for $P_T$), indicating that $\texttt{Grad-Transformer}$ fails to recover meaningful performance at this scale gap. We attribute this to the limited capacity of the current encoder-decoder backbone (Flan-T5-Large), which must map a sequence of $L_S$ block-wise update vectors to $L_T$ blocks where $L_T$ is now substantially larger (e.g., 64 layers for 32B vs. 40 layers for 14B). The output parameter space per block ($d_T$) also increases significantly, making the output projection $W_{out}$ less effective at reconstructing high-dimensional block-wise updates from a fixed-size hidden state.
>
> Importantly, this failure mode does not appear at the 14B or below scale: as shown in Table 1, all TinyLM sizes (0.5B, 1.5B, 3B) successfully generate competitive updates for 14B targets (avg. 67.73% vs. 68.04% for $P_T$). This suggests the bottleneck is the backbone Flan-T5-Large rather than a fundamental limitation of $\texttt{Grad-Transformer}$. Scaling the encoder-decoder (e.g., using Flan-T5-XL or a larger model) is a natural step that we plan to investigate in future work to generate meaningful updates for larger LLMs, such as 32B models and beyond.
>
> [1] Gao et al., Global convergence in training large-scale transformers. NeurIPS 2024.
>
> [2] Huang et al., Non-asymptotic convergence of training transformers for next-token prediction. NeurIPS 2024.
>
> [3] Izzo et al., Quantitative bounds for length generalization in transformers. arXiv preprint arXiv:2510.27015, 2025.

---

> > ### Author Rebuttal · Reviewer_rC2f · 2026-03-31
> >
> > Thank you for your response. Most of my concerns have been addressed.

---

### Decision · Program_Chairs · 2026-04-30

**Decision:**

Accept (regular)

**Comment:**

This paper addresses an important and practical problem: privacy-preserving LLM adaptation without sharing private data or requiring clients to fine-tune large models directly. Reviewers found the core idea clear and meaningful, and appreciated the scalable update-vector formulation together with the theoretical and empirical support.

The rebuttal substantially strengthened the paper. The authors addressed the main concerns with additional analysis on failure cases, segmentation strategy, privacy, and robustness under distribution shift, and several reviewers explicitly stated that their concerns were fully resolved.

Although existing some limitation such as dependence on shadow data alignment and weaker performance at larger scale gaps, these do not outweigh the paper’s strengths. Overall, I find the contribution solid and useful, and hence an acceptance.